# On the inference of complex phylogenetic networks by Markov Chain Monte-Carlo

**Charles-Elie Rabier**[1,2,3]*, **Vincent Berry**[2], **Marnus Stoltz**[1], **João D. Santos**[4,5], **Wensheng Wang**[6], **Jean-Christophe Glaszmann**[4,5], **Fabio Pardi**[2], **Celine Scornavacca**[1]*

**1** Institut des Sciences de l'Evolution (ISEM), Université de Montpellier, CNRS, EPHE, IRD, Montpellier, France, **2** Laboratoire d'Informatique, de Robotique et de Microélectronique de Montpellier (LIRMM), Université de Montpellier, CNRS, Montpellier, France, **3** Institut Montpelliérain Alexander Grothendieck (IMAG), Université de Montpellier, CNRS, Montpellier, France, **4** CIRAD, UMR AGAP, Montpellier, France, **5** Amélioration Génétique et Adaptation des Plantes méditerranéennes et tropicales (AGAP), Université de Montpellier, CIRAD, INRAE, Institut Agro, Montpellier, France, **6** Institute of Crop Sciences (ICS), Chinese Academy of Agricultural Sciences, Beijing, China

* charles-elie.rabier@umontpellier.fr (C-ER); celine.scornavacca@umontpellier.fr (CS)

**Data Availability Statement:** All data files are available from the github public repository located at https://github.com/rabier/MySnappNet.

## Abstract

For various species, high quality sequences and complete genomes are nowadays available for many individuals. This makes data analysis challenging, as methods need not only to be accurate, but also time efficient given the tremendous amount of data to process. In this article, we introduce an efficient method to infer the evolutionary history of individuals under the multispecies coalescent model in networks (MSNC). Phylogenetic networks are an extension of phylogenetic trees that can contain *reticulate* nodes, which allow to model complex biological events such as horizontal gene transfer, hybridization and introgression. We present a novel way to compute the likelihood of *biallelic* markers sampled along genomes whose evolution involved such events. This likelihood computation is at the heart of a Bayesian network inference method called SNAPPNET, as it extends the SNAPP method inferring evolutionary trees under the multispecies coalescent model, to networks. SNAPPNET is available as a package of the well-known BEAST 2 software.

Recently, the `MCMC_BiMarkers` method, implemented in PhyloNet, also extended SNAPP to networks. Both methods take biallelic markers as input, rely on the same model of evolution and sample networks in a Bayesian framework, though using different methods for computing priors. However, SNAPPNET relies on algorithms that are exponentially more time-efficient on non-trivial networks. Using simulations, we compare performances of SNAPPNET and `MCMC_BiMarkers`. We show that both methods enjoy similar abilities to recover simple networks, but SNAPPNET is more accurate than `MCMC_BiMarkers` on more complex network scenarios. Also, on complex networks, SNAPPNET is found to be extremely faster than `MCMC_BiMarkers` in terms of time required for the likelihood computation. We finally illustrate SNAPPNET performances on a rice data set. SNAPPNET infers a scenario that is consistent with previous results and provides additional understanding of rice evolution.

**Funding:** C.E.R., C.S., J.G., J.S., V.B. were partially funded by the Genome Harvest project (reference ID 1504-006, Labex Agro: ANR-10-LABX-0001-01) for study design, data collection, data analysis. C.E.R., C.S. and V.B. were also partially funded by a KIM Data & Life Sciences project (I-SITE MUSE: ANR-16-IDEX-0006) for data collection, data analysis. C.S. and M.S. were funded by the French Agence Nationale de la Recherche, through the CoCoAlSeq project (ANR-19-CE45-0012) for study design, data collection, data analysis. C.E.R., F.P. and V.B. were funded by the ATGC bioinformatic platform, a member of both the "France Génomique"network (ANR-10-INBS-0009) and the Institut Français de Bioinformatique (ANR-11-INBS-0013) for data collection and data analysis. C.E.R., C.S., M.S. and V.B. were funded by the French Agence Nationale de la Recherche, "Investissements d'Avenir" program, through the Montpellier Bioinformatics Biodiversity platform supported by the LabEx CeMEB (ANR-10-LABX-04-01) for data collection and data analysis. C.E.R. was also funded by the High Performance Computing Platform MESO@LR, financed by the Occitanie /Pyrénées-Méditerranée Region, Montpellier Mediterranean Metropole and the University of Montpellier for data collection and data analysis. C.E.R., J.G. and V.B. were also funded by the CIRAD - UMR AGAP HPC Data Center of the South Green Bioinformatics platform (http://www.south-green.fr/). for data collection and data analysis. J.G. was also funded by the French Agence Nationale de la Recherche, "Investissements d'Avenir" program, through the AdaptGrass project (reference ID170544IA, I-SITE MUSE: ANR-16-IDEX-0006) for study design, data collection and data analysis. W.W. was funded by the CGIAR Research Program on Rice Agrifood Systems (RICE) for study design and data collection.

**Competing interests:** The authors have declared that no competing interests exist.

## Author summary

Nowadays, to make the best use of the vast amount of genomic data at our disposal, there is a real need for methods able to model complex biological mechanisms such as hybridization and introgression. Understanding such mechanisms can help geneticists to elaborate strategies in crop improvement that may help reducing poverty and dealing with climate change. However, reconstructing such evolution scenarios is challenging. Indeed, the inference of phylogenetic networks, which explicitly model reticulation events such as hybridization and introgression, requires high computational resources. Thus, on large data sets, biologists generally deduce reticulation events indirectly using species tree inference tools.

In this context, we present a new Bayesian method, called SNAPPNET, dedicated to phylogenetic network inference. Our method is competitive in terms of execution speed with respect to its competitors. This speed gain enables us to consider more complex evolution scenarios during Bayesian analyses. When applied to rice genomic data, SNAPPNET retrieved an evolution scenario that confirms the global triple foundation of the species and the origin of *circum* Basmati as a hybrid derivative between Japonica cultivars and a local Indian form. It suggests that this hybridization is ancient and probably precedes the domestication of *circum* Aus.

This is a *PLOS Computational Biology* Methods paper.

## Introduction

Complete genomes for numerous species in various life domains [1–5], and even for several individuals for some species [6, 7] are nowadays available thanks to next generation sequencing. This flow of data finds applications in various fields such as pathogenecity [8], crop improvement [9], evolutionary genetics [10] or population migration and history [11–13]. Generally, phylogenomic studies use as input thousands to millions genomic fragments sampled across different species. To process such a large amount of data, methods need not only to be accurate, but also time efficient. The availability of numerous genomes at both the intra and inter species levels has been a fertile ground for studies at the interface of population genetics and phylogenetics [14] that aim to estimate the evolutionary history of closely related species. In particular, the well-known coalescent model from population genetics [15] has been extended to the *multispecies coalescent* (MSC) model [16, 17] to handle studies involving populations or individuals from several species. Recent works show how to incorporate sequence evolution processes into the MSC [18, 19]. As a result, it is now possible to reconstruct evolutionary histories while accounting for both incomplete lineage sorting (ILS) and sequence evolution [20, 21].

For a given locus, ILS leads different individuals in a same population to have different alleles that can trace back to different ancestors. Then, if speciation occurs before the different alleles get sorted in the population, the locus tree topology can differ from the species history [22]. But incongruence between these trees can also result from biological phenomena that can cause a species to inherit lineages and/or genomic fragments from more than one parent species. Examples of such phenomena include hybrid speciation [23–26], introgression [27–29]

and horizontal gene transfer [30, 31] (the latter is not addressed in this paper). As a consequence of these reticulate events, trees are not suited to represent species history, and should be replaced by phylogenetic networks. A rooted phylogenetic network is mainly a directed acyclic graph whose internal nodes can have several children, as in trees, but can also have several parents [32–34]. Various models of phylogenetic network have been proposed over time to explicitly represent reticulate evolution, such as hybridization networks [35] or ancestral recombination graphs [36], along with dozens of inference methods [37, 38].

Model-based methods have been proposed to handle simultaneously ILS and reticulate evolution, which is a desired feature to avoid bias in the inference [39–41]. These methods postulate a probabilistic model of evolution and then estimate its parameters –including the underlying network– from the data. The estimation of parameters such as branch lengths (hence speciation dates) and population sizes makes them more versatile than combinatorial methods [42]. On the down side, they usually involve high running times as they explore large parameter spaces. Two probabilistic models differentiate regarding the way a locus tree can be embedded within a network. In Kubatko's model [43, 44], all lineages of a given locus tree coalesce within a single species tree *displayed* by the network. The model of Yu et al. [45] is more general as, at each reticulation node, a lineage of the locus tree is allowed to descend from a parental ancestor independently of which ancestors provide the other lineages. Works on the latter model extend in various ways the MSC model to consider network-like evolution, giving rise to the *multispecies network coalescent* (MSNC), intensively studied in recent years [38, 41, 46–55]. For this model, Yu et al. have shown how to compute the probability of a non-recombinant locus (*gene*) tree evolving inside a network, given the branch lengths and inheritance probabilities at each reticulation node of the network [46, 48]. This opened the way to infer networks according to the well-known maximum likelihood and Bayesian statistical frameworks.

When the input data consists of multi-locus alignments, a first idea is to decompose the inference process in two steps: first, infer locus trees from their respective alignments, then look for networks that assign high probability to these trees. Following this principle, Yu et al. devised a maximum likelihood method [48], then a Bayesian sampling technique [51]. However, using locus trees as a proxy for molecular sequences loses some information contained in the alignments [16] and is subject to tree reconstruction errors. For this reasons, recent work considers jointly estimating the locus trees and the underlying network. This brings the extra advantage that better locus trees are likely to be obtained [56], but running time may become prohibitive already for inferences on few species. Wen et al. in the PHYLONET software [52] and Zhang et al. with the SPECIESNETWORK method [53] both proposed Bayesian methods following this principle.

Though a number of trees for a same locus are considered during such inference processes, they are still considered one at a time, which may lead to a precision loss (and a time loss) compared to an inference process that would consider all possible trees for a given locus at once. When data consists of a set of *biallelic* markers (e.g., SNPs), the ground-breaking work of Bryant et al. [19] allows to compute likelihoods while integrating over all gene trees, under the MSC model (*i.e.*, when representing the history as a tree). This work was recently extended to the MSNC context by Zhu et al [54].

In this paper, we present a novel way to compute the probability of biallelic markers, given a network. This likelihood computation is at the heart of a Bayesian network inference method we called SNAPPNET, as it extends the SNAPP method [19] to networks. SNAPPNET is available at https://github.com/rabier/MySnappNet and distributed as a package of the well-known BEAST 2 software [57, 58]. This package partly relies on code from SNAPP [19] to handle sequence

evolution and on code from SPECIESNETWORK [53] to modify the network during the MCMC as well as to compute network priors.

Our approach differs from that of Zhang et al. [53] in that SNAPPNET takes a matrix of biallelic markers as input while SPECIESNETWORK expects a set of alignments. Thus, the substitution models differ, as we consider only two states (alleles) while SPECIESNETWORK deals with nucleotides. The computational approaches also differ as our MCMC integrates over all locus trees for each sampled network, while SPECIESNETWORK jointly samples networks and gene trees. Though summarizing the alignments by gene trees might be less flexible, this allows SPECIESNETWORK to provide embeddings of the gene trees into the sampled networks, while in our approach this needs to be done in a complementary step after running SNAPPNET. However, managing the embeddings can also lead to computational issues as Zhang et al. report, since a topological change for the network usually requires a recomputation of the embeddings for all gene trees [53].

The SNAPPNET method we present here is much closer to the `MCMC_BiMarkers` method of Zhu et al. [54], which also extends the SNAPP method [19] to network inference. Both methods take biallelic markers as input, rely on the same model of evolution and both sample networks in a Bayesian framework. However, they differ in two important respects: the way the Bayesian inference is conducted and, most importantly, in the algorithm to compute the likelihoods. The results we present here show that this often leads to tremendous differences in running time, but also to differences in convergence.

We note that reducing running times of model-based methods can also be done by approximating likelihoods, as done by *pseudo-likelihood* methods: the network likelihood is computed for subparts of its topology, these values being then assembled to approximate the likelihood of the full network. A decomposition of the network into rooted networks on three taxa (trinets) is proposed in the PHYLONET software [49, 59] and one into semi-directed networks on four taxa in the SNaQ method of the PHYLONETWORK package [50]. Since pseudo-likelihood methods are approximate heuristics to compute a likelihood, they are usually much faster than full likelihood methods and can handle large genomic data sets. On the downside, these methods face, more often than the full-likelihood methods, serious identifiability problems since some networks simply cannot be recovered from topological substructures such as rooted triples, quartets or even embedded trees [49, 50, 60]. Here we focus on the *exact* computation of the *full* likelihood, for which identifiability issues are likely to be less serious [41, 61].

In the following, we first detail the mathematical model considered, then explain the SNAPPNET method, before illustrating its performances on simulated and real data.

## Materials and methods

### Input data

SNAPPNET considers as input data a matrix $D$ containing an alignment of $m$ biallelic markers sampled from a number of individuals. Each individual belongs to a given species. These species are in a 1-to-1 correspondence with the leaves of an unknown phylogenetic network, which is the main parameter that we wish to estimate. The markers can be SNPs or random sites sampled from chromosomes, including invariant sites. All markers are considered to be independent, so a certain distance must be preserved between genomic locations included in the matrix. We identify the two alleles with the red and green colors.

Each column $D_i$ of the alignment corresponds to a different marker. The only information that is relevant to SNAPPNET's computations are the numbers of red and green alleles observed in $D_i$ for the individuals of a given species. This implies that unphased data can be analyzed with SNAPPNET, as long as it is translated in the input format expected by the software.

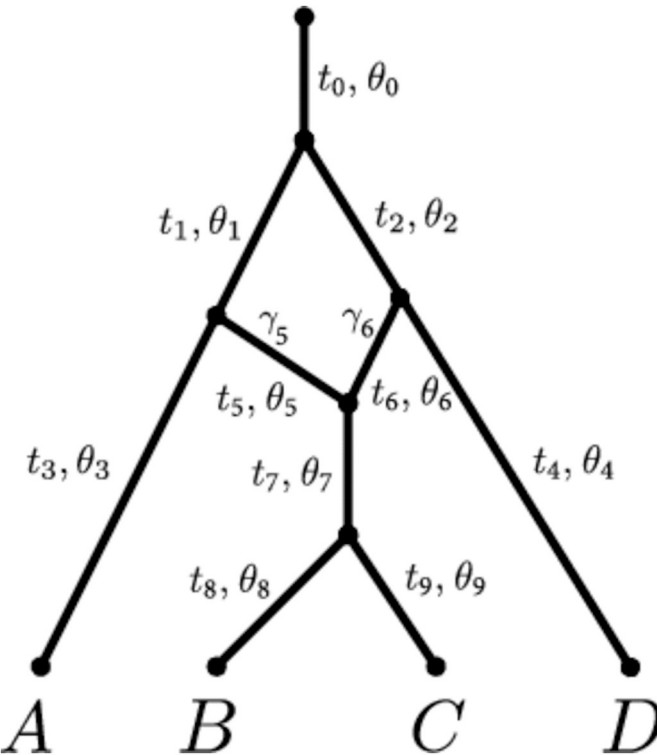

**Fig 1. Example of a phylogenetic network.** The top node represents the origin and its child node is called the root of the network. Time flows from the origin node to the leaves (here *A*, *B*, *C*, *D*) so branches are directed from the top to the leaves. Each branch *x* is associated to a length $t_x$, and to a population size $\theta_x$. Additionally, branches *x* on top of a reticulation node have an inheritance probability $\gamma_x$ representing their probability to have contributed to any individual at the top of the branch just below.

## Mathematical model

In this paper, we refer to phylogenetic networks as directed acyclic graphs with branches oriented as the time flows, see Fig 1. At their extremities, networks have a single node with no incoming branch and a single outgoing branch —the *origin*— and a number of nodes with a single incoming branch and no outgoing branches —the *leaves*. All other nodes either have a single incoming branch and two outgoing branches —the *tree* nodes— or two incoming branches and a single outgoing branch —the *reticulation* nodes. Tree nodes and reticulation nodes represent speciations and hybridization events, respectively. For consistency with Zhang et al. [53], the immediate descendant of the origin—that is, the tree node representing the first speciation in the network—is called the *root*.

Each branch *x* in the network represents a population, and is associated to two parameters: a scaled population size $\theta_x$ and a branch length $t_x$. Any branch *x* on top of a reticulation node *h* is further associated with a probability $\gamma_x \in (0, 1)$, under the constraint that the probabilities of the two parent branches of *h* sum to 1. These probabilities are called *inheritance probabilities*. All these parameters have a role in determining how gene trees are generated by the model, and how markers evolve along these gene trees, as described in the next two subsections, respectively.

**Gene tree model.** Gene trees are obtained according to the MSNC model. The process starts at the leaves of the network, where a given number of lineages is sampled for each leaf,

each lineage going backwards in time, until all lineages coalesce. Along the way, this process determines a gene tree whose branch lengths are each determined as the amount of time between two coalescences affecting a same lineage. Here and in what follows, "times" —and therefore branch lengths— are always measured in terms of expected number of mutations per site.

Within each branch $x$ of the network, the model applies a standard coalescent process governed by $\theta_x$. In detail, any two lineages within $x$ coalesce at rate $2/\theta_x$, meaning that the first coalescent time among $k$ lineages follows an exponential distribution $\mathcal{E}(k(k-1)/\theta_x)$, since the coalescence of each combination of 2 lineages is equiprobable. Naturally, if the waiting time to coalescence exceeds the branch length $t_x$, the lineages are passed to the network branch(es) above $x$ without coalescence. If there are two such branches, say $y$, $z$ (i.e., the origin of $x$ is a reticulation node) then each lineage that has arrived at the top of branch $x$ chooses independently whether it goes to $y$ or $z$ with probabilities $\gamma_y$ and $\gamma_z = 1 - \gamma_y$, respectively [45]. The process terminates when all lineages have coalesced and only one ancestral lineage remains.

**Mutation model.**   As is customary for unlinked loci, we assume that the data is generated by a different gene tree for each biallelic marker. The evolution of a marker along the branches of this gene tree follows a two-states asymmetric continuous-time Markov model, scaled so as to ensure that 1 mutation is expected per time unit. This is the same model as Bryant et al. [19]. For completeness, we describe this mutation model below.

We represent the two alleles by red and green colors. Let $u$ and $v$ denote the instantaneous rates of mutating from red to green, and from green to red, respectively. Then, for a single lineage, $\mathbb{P}(\text{red at } t + \Delta t \mid \text{green at } t) = v\Delta t + o(\Delta t)$, and $\mathbb{P}(\text{green at } t + \Delta t \mid \text{red at } t) = u\Delta t + o(\Delta t)$, where $o(\Delta t)$ is negligible when $\Delta t$ tends to zero. The stationary distribution for the allele at the root of the gene tree is green with probability $u/(u + v)$ and red with probability $v/(u + v)$. Under this model, the expected number of mutations per time unit is $2uv/(u + v)$. In order to measure time (branch lengths) in terms of expected mutations per site (i.e. genetic distance), we impose the constraint $2uv/(u + v) = 1$ as in [19]. When $u$ and $v$ are set to 1, the model is also known as the Haldane model [62] or the Cavender-Farris-Neyman model [63].

## Bayesian framework

**Posterior distribution.**   Let $D_i$ be the data for the $i$-th marker. The posterior distribution of the phylogenetic network $\Psi$ can be expressed as:

$$\begin{aligned}
\mathbb{P}(\Psi|D_1, \ldots, D_m) \quad &\propto \mathbb{P}(D_1, \ldots, D_m \mid \Psi) \cdot \mathbb{P}(\Psi) \\
&= \mathbb{P}(\Psi) \cdot \prod_{i=1}^{m} \mathbb{P}(D_i|\Psi)
\end{aligned} \quad (1)$$

where $\propto$ means "is proportional to", and where $\mathbb{P}(D_1, \ldots, D_m \mid \Psi)$ and $\mathbb{P}(\Psi)$ refer to the likelihood and the network prior, respectively.

Eq 1—which relies on the independence of the data at different markers— allows us to compute a quantity proportional to the posterior by only using the prior of $\Psi$ and the likelihoods of $\Psi$ with respect to each marker, that is $\mathbb{P}(D_i|\Psi)$. While we could approximate $\mathbb{P}(D_i|\Psi)$ by sampling gene trees from the distribution determined by the species network, this is time-consuming and not necessary. Similarly to the work by Bryant et al. [19] for inferring phylogenetic *trees*, we show below that $\mathbb{P}(D_i|\Psi)$ can be computed for *networks* using dynamic programming.

SNAPPNET samples networks from their posterior distribution by using Markov chain Monte-Carlo (MCMC) based on Eq 1.

**Priors.**   Before describing the network prior, let us recall the network components: the topology, the branch lengths, the inheritance probabilities and the populations sizes. In this context, we used the birth-hybridization process of Zhang et al. [53] to model the network topology and its branch lengths. This process depends on the speciation rate λ, on the hybridization rate nu and on the time of origin $\tau_0$. Hyperpriors are imposed onto these parameters. An exponential distribution is used for the hyperparameters $d := \lambda - $ nu and $\tau_0$. The hyperparameter $r :=$ nu$/\lambda$ is assigned a Beta distribution. We refer to [53] for more details. The inheritance probabilities are modeled according to a uniform distribution. Moreover, like SNAPP, SNAPPNET considers independent and identically distributed Gamma distributions as priors on population sizes $\theta_x$ associated to each network branch. This prior on each population size induces a prior on the corresponding coalescence rate (see [19] and SNAPP's code). Last, as in SNAPP, the user can specify fixed values for the $u$ and $v$ mutation rates, or impose a prior for these rates and let them be sampled within the MCMC.

**Partial likelihoods.**   In the next section we describe a few recursive formulae that we use to calculate the likelihood $\mathbb{P}(D_i|\Psi)$ using a dynamic programming algorithm. Here we introduce the notations that allow us to define the quantities involved in our computations. Unless otherwise stated, notations that follow are relative to the $i$th biallelic marker. To keep the notations light, the dependence on $i$ is not explicit.

Given a branch $x$, we denote by $\overline{x}$ and $\underline{x}$ the top and bottom of that branch. We call $\overline{x}$ and $\underline{x}$ *population interfaces*. We say that two population interfaces are *incomparable* if neither is a descendant of the other (which also excludes them being equal). $N_{\overline{x}}$ and $N_{\underline{x}}$ are random variables denoting the number of gene tree lineages at the top and at the bottom of $x$, respectively. Similarly, $R_{\overline{x}}$ and $R_{\underline{x}}$ denote the number of red lineages at the top and bottom of $x$, respectively. See Fig 2 for illustration of these concepts and of the notation that we introduce in the following.

For simplicity, when $x$ is a branch incident to a leaf, we identify $\underline{x}$ with that leaf. Two quantities that are known about each leaf are $r_{\underline{x}}$ and $n_{\underline{x}}$, which denote the number of red lineages sampled at $\underline{x}$ and the total number of lineages sampled at $\underline{x}$, respectively. Note that $N_{\underline{x}}$, in this case, is non-random: indeed, it must necessarily equal $n_{\underline{x}}$, which is determined by the number of individuals sampled from that species. On the other hand, the model we adopt determines a distribution for the $R_{\underline{x}}$. The probability of the observed values $r_{\underline{x}}$ for these random variables equals $\mathbb{P}(D_i|\Psi)$.

Now let **x** be an ordered collection (i.e. a vector) of population interfaces. We use $\mathbf{n_x}$ (or $\mathbf{r_x}$) to denote a vector of non-negative integers in a 1-to-1 correspondence with the elements of **x**. Then $N_\mathbf{x} = \mathbf{n_x}$ is a shorthand for the equations expressing that the numbers of lineages in $\mathbf{n_x}$ are observed at the respective interfaces in **x**. For example, if $\mathbf{x} = (\underline{x}, \overline{y})$ and $\mathbf{n_x} = (m, n)$, then $N_\mathbf{x} = \mathbf{n_x}$ is a shorthand for $N_{\underline{x}} = m, N_{\overline{y}} = n$. We use $R_\mathbf{x} = \mathbf{r_x}$ analogously to express the observation of the numbers of red lineages in $\mathbf{r_x}$ at **x**.

In order to calculate the likelihood $\mathbb{P}(D_i|\Psi)$, we subdivide the problem into that of calculating quantities that are analogous to partial likelihoods. Given a vector of population interfaces **x**, let $\mathbf{L(x)}$ denote a vector containing the leaves that descend from any element of **x**, and let $\mathbf{r_{L(x)}}$ be the vector containing the numbers of red lineages $r_{\underline{x}}$ observed at each leaf $\underline{x}$ in $\mathbf{L(x)}$. Then we define:

$$\mathbf{F_x}(\mathbf{n_x}; \mathbf{r_x}) = \mathbb{P}(R_{\mathbf{L(x)}} = \mathbf{r_{L(x)}} \mid N_\mathbf{x} = \mathbf{n_x}, R_\mathbf{x} = \mathbf{r_x}) \cdot \mathbb{P}(N_\mathbf{x} = \mathbf{n_x}) \qquad (2)$$

(see Fig 2). These quantities are generalizations of similar quantities defined by Bryant et al.

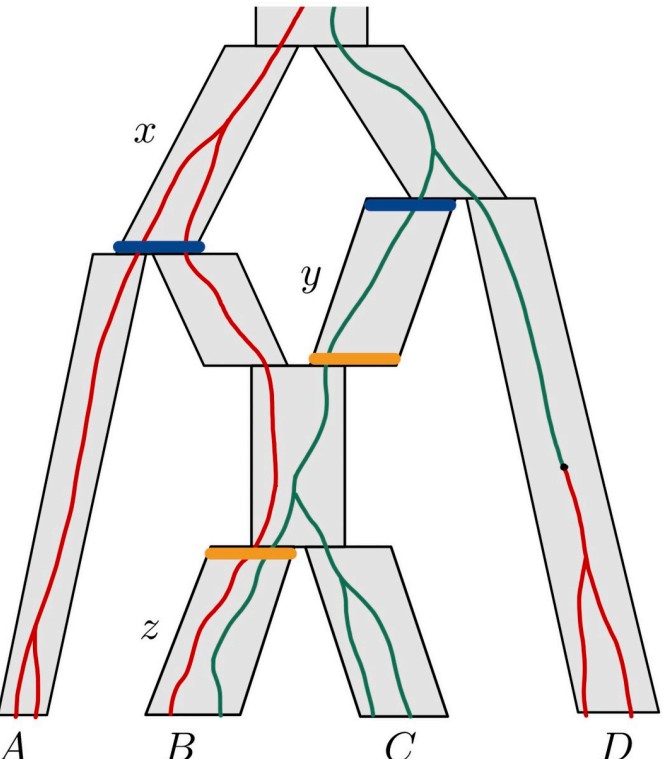

**Fig 2. Illustration of the concepts and notation employed to describe likelihood computations.** The species network topology is the same as that in Fig 1, but branches (populations) are now represented as grey parallelograms. A gene tree is drawn inside the species network (green and red lines). One mutation occurs in the branch above $D$. We focus on three branches: $x$, $y$ and $z$. Colored horizontal bars represent the population interfaces $\underline{x}$, $\overline{y}$, $\underline{y}$ and $\overline{z}$. Note that $(\underline{x}, \overline{y})$ (blue) is a vector of incomparable population interfaces, while $(\underline{y}, \overline{z})$ (orange) is not, as $\overline{z}$ is a descendant of $\underline{y}$. Here, $n_A = n_B = n_C = n_D = 2$, $r_A = 2$, $r_B = 1$, $r_C = 0$, $r_D = 2$ are known, whereas the values of $N_{\underline{x}}, N_{\overline{y}}, N_{\underline{y}}, N_{\overline{z}}$ and $R_{\underline{x}}, R_{\overline{y}}, R_{\underline{y}}, R_{\overline{z}}$ are not observed, and depend on the gene tree generated by the MSNC process. For the gene tree shown, $N_{(\underline{x},\overline{y})} = (2, 1)$ and $R_{(\underline{x},\overline{y})} = (2, 0)$. Since $z$ is incident to leaf $B$, we have $\underline{z} = B$ and $R_{\underline{z}} = r_B = 1$. Now note $\mathbf{L}((\underline{x}, \overline{y})) = (A, B, C)$. Then, $\mathbf{F}_{(\underline{x},\overline{y})}((n, n'); (r, r')) = \mathbb{P}(R_A = r_A, R_B = r_B, R_C = r_C \mid N_{\underline{x}} = n, N_{\overline{y}} = n', R_{\underline{x}} = r, R_{\overline{y}} = r')\mathbb{P}(N_{\underline{x}} = n, N_{\overline{y}} = n')$.

[19]. We will call them partial likelihoods, although, as noted by these authors, strictly speaking this is an abuse of language.

**Computing partial likelihoods: The rules.** Here we show a set of rules that can be applied to compute partial likelihoods in a recursive way. Derivations and detailed proofs of the correctness of these rules can be found in Section 1 in S1 Text.

We use the following conventions. In all the rules that follow, vectors of population interfaces **x,y,z** are allowed to be empty. The comma operator is used to concatenate vectors or append new elements at the end of vectors, for example, if $\mathbf{a} = (a_1, a_2, \ldots, a_k)$ and $\mathbf{b} = (b_1, b_2, \ldots, b_h)$, then $\mathbf{a}, \mathbf{b} = (a_1, \ldots, a_k, b_1, \ldots, b_h)$ and $\mathbf{a}, c = (a_1, a_2, \ldots, a_k, c)$. Trivially, if $\mathbf{a}$ is empty, then $\mathbf{a}, \mathbf{b} = \mathbf{b}$ and $\mathbf{a}, c = (c)$. A vector **x** of incomparable population interfaces is one where all pairs of population interfaces are incomparable. Finally, for any branch $x$, let $m_x$ denote the number of lineages sampled in the descendant leaves of $x$.

**Rule 0:** Let $x$ be a branch incident to a leaf. Then,

$$\mathbf{F}_{(\underline{x})}((n); (r)) = \mathbb{1}\{n = n_{\underline{x}}\} \cdot \mathbb{1}\{r = r_{\underline{x}}\}$$

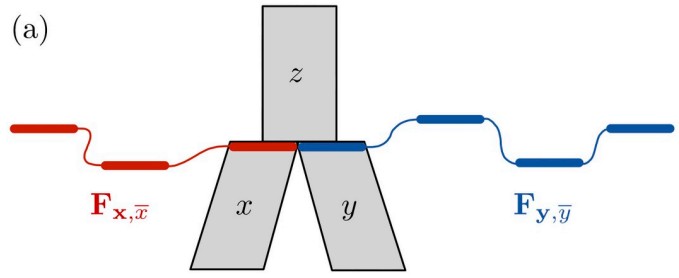

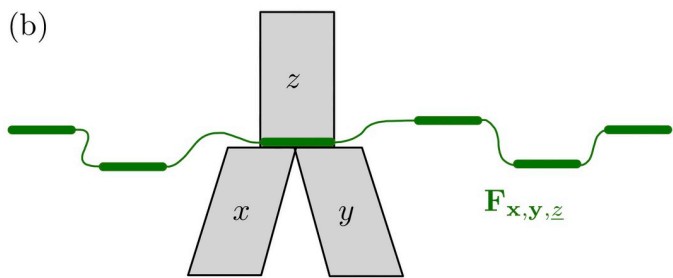

**Fig 3. Illustration of Rule 2.** Given (a) the partial likelihoods for the $\mathbf{x}, \overline{x}$ (red) vector of population interfaces and the partial likelihoods for the $\mathbf{y}, \overline{y}$ (blue) vector of population interfaces, Rule 2 allows us to compute the partial likelihoods for the (green) vector $\mathbf{x}, \mathbf{y}, \underline{z}$ (b).

**Rule 1:** Let $\mathbf{x}, \underline{x}$ be a vector of incomparable population interfaces. Then,

$$\mathbf{F}_{\mathbf{x},\overline{x}}(\mathbf{n}_{\mathbf{x}}, n_{\overline{x}}; \mathbf{r}_{\mathbf{x}}, r_{\overline{x}}) = \sum_{n=n_{\overline{x}}}^{m_x} \sum_{r=0}^{n} \mathbf{F}_{\mathbf{x},\underline{x}}(\mathbf{n}_{\mathbf{x}}, n; \mathbf{r}_{\mathbf{x}}, r) \exp\left(\mathbb{Q}_x t_x\right)_{(n,r);(n_{\overline{x}}, r_{\overline{x}})}$$

where $t_x$ denotes the length of branch $x$, and $\mathbb{Q}_x$ is the rate matrix defined by Bryant et al. [19, p. 1922] that accounts for both coalescence and mutation (see also Section 1 in S1 Text).

**Rule 2:** Let $\mathbf{x}, \overline{x}$ and $\mathbf{y}, \overline{y}$ be two vectors of incomparable population interfaces, such that $\mathbf{L}(\mathbf{x}, \overline{x})$ and $\mathbf{L}(\mathbf{y}, \overline{y})$ have no leaf in common. If $x$ and $y$ are the immediate descendants of a branch $z$, as in Fig 3, then,

$$\mathbf{F}_{\mathbf{x},\mathbf{y},\underline{z}}(\mathbf{n}_{\mathbf{x}}, \mathbf{n}_{\mathbf{y}}, n_{\underline{z}}; \mathbf{r}_{\mathbf{x}}, \mathbf{r}_{\mathbf{y}}, r_{\underline{z}}) =$$
$$\sum_{n_{\overline{x}}} \sum_{r_{\overline{x}}} \mathbf{F}_{\mathbf{x},\overline{x}}(\mathbf{n}_{\mathbf{x}}, n_{\overline{x}}; \mathbf{r}_{\mathbf{x}}, r_{\overline{x}}) \mathbf{F}_{\mathbf{y},\overline{y}}(\mathbf{n}_{\mathbf{y}}, n_{\underline{z}} - n_{\overline{x}}; \mathbf{r}_{\mathbf{y}}, r_{\underline{z}} - r_{\overline{x}}) \binom{n_{\overline{x}}}{r_{\overline{x}}} \binom{n_{\underline{z}} - n_{\overline{x}}}{r_{\underline{z}} - r_{\overline{x}}} \binom{n_{\underline{z}}}{r_{\underline{z}}}^{-1}$$

The ranges of $n_{\overline{x}}$ and $r_{\overline{x}}$ in the summation terms are defined by $\max(0, n_{\underline{z}} - m_y) \le n_{\overline{x}} \le \min(m_x, n_{\underline{z}})$ and $\max(0, n_{\overline{x}} + r_{\underline{z}} - n_{\underline{z}}) \le r_{\overline{x}} \le \min(n_{\overline{x}}, r_{\underline{z}})$.

**Rule 3:** Let $\mathbf{x}, \overline{x}$ be a vector of incomparable population interfaces, such that branch $x$'s top node is a reticulation node. Let $y, z$ be the branches immediately ancestral to $x$, as in Fig 4.

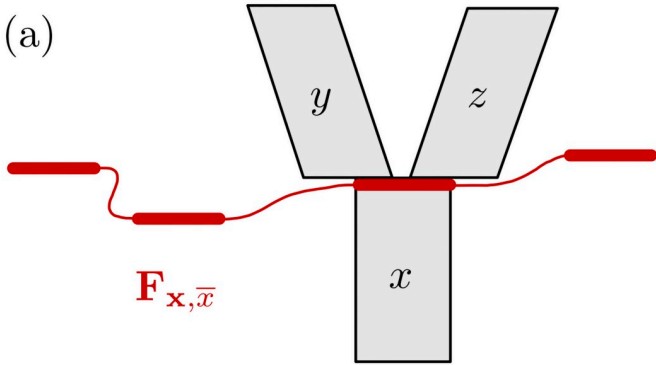

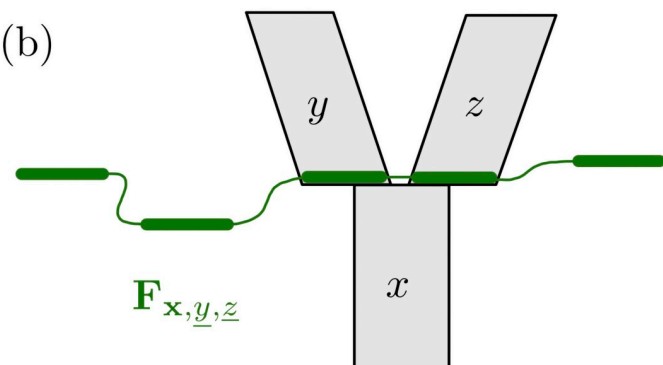

**Fig 4. Illustration of Rule 3.** Given (a) the partial likelihoods for the $\mathbf{x}, \overline{x}$ (red) vector of population interfaces, Rule 3 allows us to compute the partial likelihoods for the (green) vector $\mathbf{x}, \underline{y}, \underline{z}$ (b).

Then,

$$\mathbf{F}_{\mathbf{x},\underline{y},\underline{z}}(\mathbf{n}_{\mathbf{x}}, n_{\underline{y}}, n_{\underline{z}}; \mathbf{r}_{\mathbf{x}}, r_{\underline{y}}, r_{\underline{z}}) = \mathbf{F}_{\mathbf{x},\overline{x}}(\mathbf{n}_{\mathbf{x}}, n_{\underline{y}} + n_{\underline{z}}; \mathbf{r}_{\mathbf{x}}, r_{\underline{y}} + r_{\underline{z}}) \begin{pmatrix} n_{\underline{y}} + n_{\underline{z}} \\ n_{\underline{y}} \end{pmatrix} \gamma_{\underline{y}}^{n_{\underline{y}}} \cdot \gamma_{\underline{z}}^{n_{\underline{z}}}$$

**Rule 4:** Let $\mathbf{z}, \overline{x}, \overline{y}$ be a vector of incomparable population interfaces, and let $x, y$ be immediate descendants of branch $z$, as in Fig 5. Then,

$$\mathbf{F}_{\mathbf{z},\underline{z}}(\mathbf{n}_{\mathbf{z}}, n_{\underline{z}}; \mathbf{r}_{\mathbf{z}}, r_{\underline{z}}) =$$

$$\sum_{n_{\overline{x}}} \sum_{r_{\overline{x}}} \mathbf{F}_{\mathbf{z},\overline{x},\overline{y}}(\mathbf{n}_{\mathbf{z}}, n_{\overline{x}}, n_{\underline{z}} - n_{\overline{x}}; \mathbf{r}_{\mathbf{z}}, r_{\overline{x}}, r_{\underline{z}} - r_{\overline{x}}) \begin{pmatrix} n_{\overline{x}} \\ r_{\overline{x}} \end{pmatrix} \begin{pmatrix} n_{\underline{z}} - n_{\overline{x}} \\ r_{\underline{z}} - r_{\overline{x}} \end{pmatrix} \begin{pmatrix} n_{\underline{z}} \\ r_{\underline{z}} \end{pmatrix}^{-1}$$

The ranges of $n_{\overline{x}}$ and $r_{\overline{x}}$ in the sums are the same as those in Rule 2.

Note that, in the rules above, we assume that the vectors of population interfaces (*VPIs* from here on) on the right-hand side of each equation only contain incomparable population interfaces. This is necessary to ensure the validity of the rules (see Section 1 in S1 Text). It is

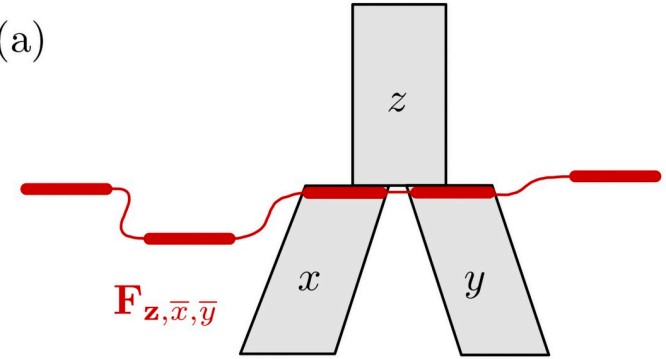

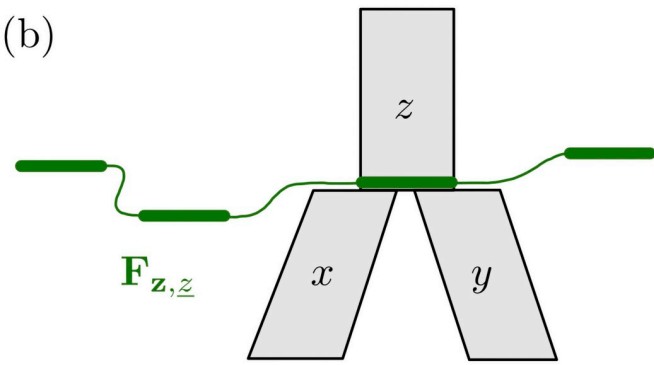

**Fig 5. Illustration of Rule 4.** Given (a) the partial likelihoods for the $\mathbf{z}, \overline{x}, \overline{y}$ (red) vector of population interfaces, Rule 4 allows us to compute the partial likelihoods for the (green) vector $\mathbf{z}, \underline{z}$ (b).

easy to verify that, as a consequence of that assumption, also the VPIs on the left-hand side of each equation only contain incomparable population interfaces. Therefore, repeated application of the rules can only result in a partial likelihood $\mathbf{F_x}(\mathbf{n_x};\mathbf{r_x})$ where $\mathbf{x}$ is a vector of incomparable population interfaces. Thus, all VPIs that we will deal with only contain incomparable population interfaces.

Repeated application of the rules above, performed by an algorithm described in the next subsection, leads eventually to the partial likelihoods for $\underline{\rho}$, the population interface immediately above the root of the network (i.e, $\rho$ is the branch linking the origin to the root). From these partial likelihoods, the full likelihood $\mathbb{P}(D_i|\Psi)$ is computed as follows:

$$\mathbb{P}(D_i \mid \Psi) = \sum_{n=1}^{m_\rho}\sum_{r=0}^{n}\mathbf{F}_{(\underline{\rho})}(n;r) \cdot \mathbb{P}(R_{\underline{\rho}} = r \mid N_{\underline{\rho}} = n), \tag{3}$$

where the conditional probabilities $\mathbb{P}(R_{\underline{\rho}} = r \mid N_{\underline{\rho}} = n)$ are obtained as described by Bryant et al. [19]. Note that the length of branch $\rho$ does not play any role in the computation of the likelihood, so it is not identifiable.

**Likelihood computation.** We now describe the algorithm that allows SNAPPNET to derive the full likelihood $\mathbb{P}(D_i|\Psi)$ using the rules introduced above. We refer to Section 2 in S1 Text for detailed pseudocode.

The central ingredient of this algorithm are the partial likelihoods for a VPI $\mathbf{x}$, which are stored in a matrix with potentially high dimension, denoted $\mathbf{F_x}$. We say that a VPI $\mathbf{x}$ is *active* at some point during the execution of the algorithm, if: (1) $\mathbf{F_x}$ has been computed by the algorithm, (2) $\mathbf{F_x}$ has not yet been used to compute the partial likelihoods for another VPI. To reduce memory usage, we only store $\mathbf{F_x}$ for active VPIs.

In a nutshell, the algorithm traverses each node in the network following a topological sort [64], that is, in an order ensuring that a node is only traversed after all its descendants have been traversed. Every node traversal involves deriving the partial likelihoods of a newly active VPI from those of at most two VPIs that, as a result, become inactive. Eventually, the root of the network is traversed, at which point the only active VPI is $(\underline{\rho})$ and the full likelihood of the network is computed from $\mathbf{F}_{(\underline{\rho})}$ using Eq 3.

In more detail, a node is ready to be traversed when all its child nodes have been traversed. At the beginning, only leaves can be traversed and their partial likelihoods $\mathbf{F}_{(\underline{x})}$ are obtained by application of Rule 0, followed by Rule 1 to obtain $\mathbf{F}_{(\overline{x})}$. Every subsequent traversal of a node $d$ entails application of one rule among Rules 2, 3 or 4, depending on whether $d$ is a tree node and on whether the branch(es) topped by $d$ correspond to more than one VPI (see Figs 3–5). The selected rule computes $\mathbf{F_x}$ for a newly active VPI $\mathbf{x}$. This is then followed by application of Rule 1 to replace every occurrence of any population interface $\underline{x}$ in $\mathbf{x}$ with $\overline{x}$.

It is helpful to note that at any moment, the set of active VPIs forms a frontier separating the nodes that have already been traversed, from those that have not yet been traversed (i.e., if branch $x = (d, e)$ with $d$ not traversed and $e$ traversed, then there must be an active VPI with $\underline{x}$ or $\overline{x}$ among its population interfaces). Any node that lies immediately above this frontier can be the next one to be traversed. Thus, there is some latitude in the choice of the complete order in which nodes are traversed. Different orders will lead to different VPIs being activated by the algorithm, which in turn will lead to different running times. In fact, running times are largely determined by the sizes of the VPIs encountered. This point is explored further in the next section.

The correctness of our implementation of the algorithm above was confirmed by comparing the likelihoods we obtain to those computed with `MCMC_BiMarkers`, which also relies on biallelic marker data [54].

**Time complexity of computing the likelihood.**   Our approach improves the running times by several orders of magnitude with respect to `MCMC_BiMarkers` [54]. This is clearly apparent for some experiments detailed in the Results section, but it can also be understood by comparing computational complexities.

Here, let $n$ be the total number of individuals sampled, and let $s$ denote the size of the species network $\Psi$ (i.e. its number of branches or its number of nodes). Let us first examine the running time to process one node in $\Psi$. For any of Rules 0–4, let $K$ be the number of population interfaces in the VPI for which partial likelihoods are being computed, that is, $K$ is the number of elements of $\mathbf{x}, \overline{x}$ for Rule 1, that of $\mathbf{x}, \mathbf{y}, \underline{z}$ for Rule 2, and so on. These partial likelihoods are stored in a $2K$-dimensional matrix, with $O(n^{2K})$ elements. Each rule specifies how to compute an element of this matrix in at most $O(n^2)$ operations (in fact rules 0 and 3 only require $O(1)$ operations). Thus, any node in the network can be processed in $O(n^{2K+2})$ time.

Since the running time of any other step—i.e. computing Eq 3, and $\exp(\mathbb{Q}_x t_x)$—is dominated by these terms, the total running time is $O(sn^{2\overline{K}+2})$, where $\overline{K}$ is the maximum number of population interfaces in a VPI activated by the given traversal, and where s is the size of the species network.

Let us now compare this to the complexity of the likelihood computations described by Zhu et al. [54]. Processing a node $d$ of the network in their algorithm involves at most $O(n^{4r_d+4})$ time, where $r_d$ is the number of reticulation nodes which descend from $d$, and for

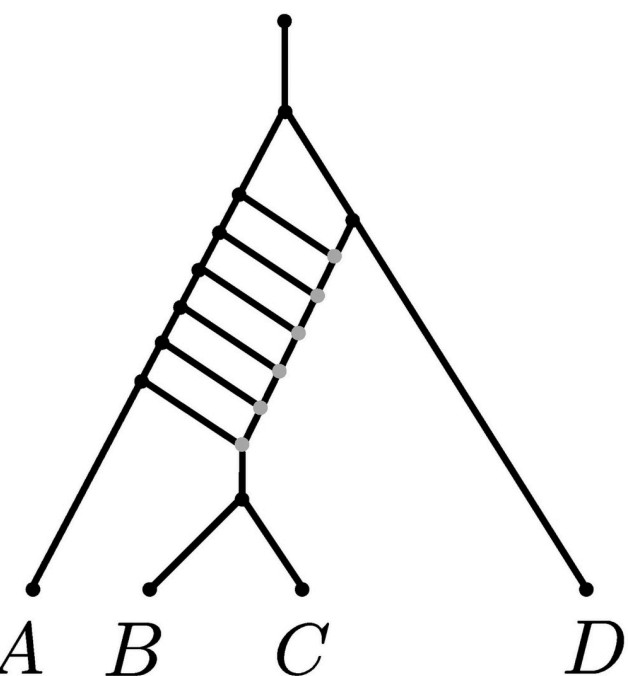

**Fig 6. Example of a phylogenetic network where the level $\ell$ is equal to 6 (the reticulation nodes are depicted in grey), while $\overline{K} \in \{3, 4, 5, 6, 7\}$, depending on the traversal algorithm (not shown).** A traversal ensuring that $\overline{K}$ remains close to the lower end of this interval (the *scanwidth* of the network [66]) will be several orders of magnitude faster than algorithms whose complexity depends exponentially on $\ell$. Increasing the number of reticulation nodes while keeping a "ladder" topology as above can make $\ell$ arbitrarily large, while the scanwidth remains constant. This topology may seem odd but it is intended as the backbone of a more complex and realistic network with subtrees hanging from the different internal branches of the ladder, in which case the complexity issue remains.

which there exists a path from $d$ that does not pass via a *lowest articulation node* (see definitions in Zhu et al. [54]). In Section 3 of S1 Text, we show that this entails a total running time of $O(sn^{4\ell+4})$, where $\ell$ is the *level* of the network [32, 65].

Thus, the improvement in running times with respect to the algorithm by Zhu et al. [54] relies on the fact that $2\overline{K} + 2 \ll 4\ell + 4$. One way of seeing this is to remark that, for any traversal of the network, $\overline{K} \leq \ell + 1$. We refer to Section 3 in S1 Text for a proof of this result. Assuming that $\overline{K}$ and $\ell$ are close, this would imply that the exponent of $n$ in the worst-case time complexity is roughly halved with respect to Zhu et al. [54]. However, $\overline{K}$ is potentially much smaller than the level $\ell$, as depicted in Fig 6.

We call the minimum value of $\overline{K}$ over all possible traversals of the network the *scanwidth* of the network [66]. The current implementation of SNAPPNET chooses an arbitrary traversal of the network, but research is ongoing to further lower running times by relying on more involved traversal algorithms producing VPIs with sizes closer to the scanwidth [66].

**MCMC operators.** SNAPPNET incorporates the MCMC operators of SPECIESNETWORK [53] to move through the network space, and also benefits from operators specific to the mathematical model behind SNAPP [19] (e.g. population sizes, mutation rates, etc.).

In order to explore the network space, we used the following topological operators from SPECIESNETWORK: (a) *addReticulation* and (b) *deleteReticulation* add and delete reticulation nodes respectively, (c) *flipReticulation* flips the direction of a reticulation branch and finally (d) *relocateBranch* and (e) *relocateBranchNarrow* relocate either the source or the destination

of random branch. The operators on gene trees from SPECIESNETWORK have been discarded since in SNAPPNET gene trees are integrated out. The following SNAPP operators acting on continuous parameters are incorporated within SNAPPNET: (a) *changeUAndV* changes the values of the instantaneous rates *u* and *v*, (b) *changeGamma* and (c) *changeAllGamma* scale a single population size or all population sizes, respectively.

Last, SNAPPNET takes also advantage of a few SPECIESNETWORK operators for continuous parameters: (a) *turnOverScale* and (b) *divRateScale* allow to change respectively the hyperparameters *r* and *d* for the birth-hybridization process, (c) *inheritanceProbUniform* and (d) *inheritanceProbRndWalk* transform the inheritance probability $\gamma$ at a random reticulation node by drawing either a uniformly distributed number or by applying a uniform sliding window to the logit of $\gamma$, (e) *networkMultiplier* and (f) *originMultiplier* scale respectively the heights of all internal nodes or of the origin node, (g) *nodeUniform* and (h) *nodeSlider* move the height of a random node uniformly or using a sliding window.

In summary, SNAPPNET relies on 16 MCMC operators, described in SNAPPNET's manual (https://github.com/rabier/MySnappNet). We refer to the original publications introducing these operators for more details [19, 53].

## Simulation study

**Simulated data.**   We implemented a simulator called SIMSNAPPNET, an extension to networks of the SIMSNAPP software [19]. SIMSNAPPNET handles the MSNC model whereas SIMSNAPP relies on the MSC model. SIMSNAPPNET is available at https://github.com/rabier/SimSnappNet. In all simulations, we considered a given phylogenetic network, and a gene tree was simulated inside the network, according to the MSNC model. Next, a Markov process was generated along the branches of the gene tree, in order to simulate the evolution of a marker. Note that markers at different sites rely on different gene trees. In all cases, we set both the *u* and *v* rates to 1. Moreover, we used the same $\theta = 0.005$ value, for all network branches. Our configuration differs slightly from the one of [54]. These authors considered $\theta = 0.006$ for external branches and $\theta = 0.005$ for internal branches. Indeed, since SNAPPNET considers the same prior distribution $\Gamma(\alpha, \beta)$ for all $\theta$'s, we found it more appropriate to generate data under SNAPPNET's assumptions.

Three numbers of markers were studied: 1,000, 10,000 or 100,000 biallelic sites were generated. Unless otherwise stated, constant sites were not discarded since SNAPPNET's mathematical formulas rely on random markers. When the analysis relied only on polymorphic sites, the gene tree and the associated marker were regenerated until it resulted in a polymorphic site. We considered 20 replicates for each simulation set up.

**Phylogenetic networks studied.**   We studied the three phylogenetic networks shown in Fig 7. Networks A and B are rather simple networks that we wish our tool to be able to infer. They have been taken from [54] and this allowed us to compare the performances of SNAPPNET and MCMC_BiMarkers on these networks, without having to rerun the latter. Networks A and B have one and two reticulations, respectively. Network C, like B, has two reticulations, but their relative positions are different: in C they are on top of one another, allowing us to investigate the influence of nested reticulations on the inference. In order to fully describe these networks, we give their extended Newick representation [67] in Section 4 in S1 Text.

We also studied networks C(3) and C(4), which are variants of network C (see Fig 8). Network C(*k*)—containing *k* reticulation nodes—is obtained by splitting species *C* into $k - 1$ species, named $C_1, C_2, \ldots, C_{k-1}$, and by adding reticulations between them in the way depicted in Fig 8. The relative positions of reticulation nodes in these networks represents a significant computational challenge for network inference tools, and were therefore used to

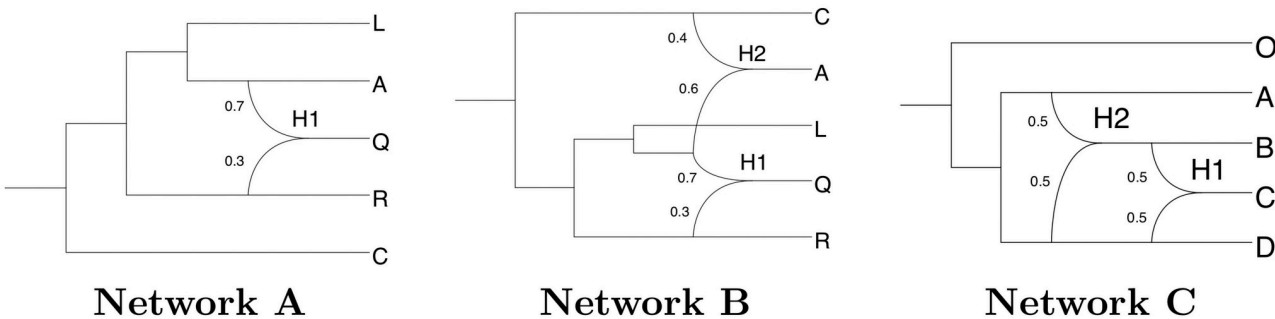

**Fig 7. The three phylogenetic networks used for simulating data.** Networks A and B are taken from [54]. Branch lengths are measured in units of expected number of mutations per site (i.e. substitutions per site). Displayed values represent inheritance probabilities.

evaluate the efficiency of a single likelihood computation performed by SNAPPNET and MCMC_BiMarkers.

**Bayesian analysis.** In the experiments on networks A, B and C, we used a single tree as initial state of the MCMC. None of the starting trees were subtrees of the correct network topology. A few alternative starting trees were used to check the convergence of the MCMC, showing a limited effect of the starting tree on the posterior probabilities. All relevant Newick representations are reported in S1 Text.

As priors on population sizes, we considered $\theta \sim \Gamma(1,200)$ for all branches. Since simulated data were generated by setting $\theta = 0.005$, the expected value of this prior distribution is exactly

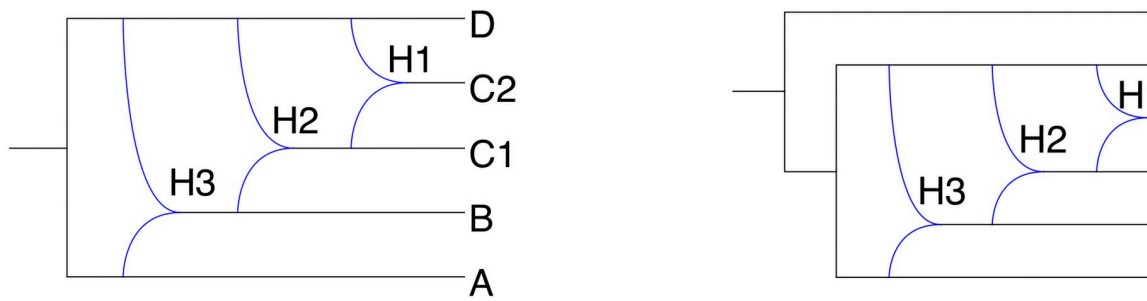

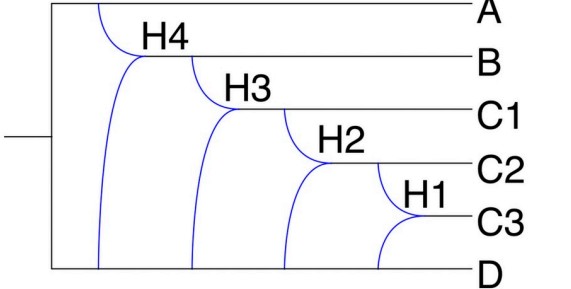 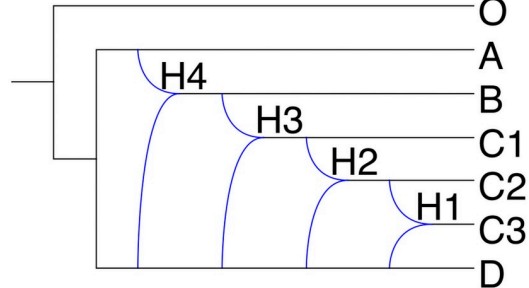

**Fig 8. The networks from the C family, with either 3 or 4 reticulation nodes, and with or without outgroup O.**

matching the true value ($\mathbb{E}(\theta) = 0.005$). For calibrating the network prior, we chose the same distributions as suggested in [53]: $d \sim \mathcal{E}(0.1)$, $r \sim \text{Beta}(1, 1)$, $\tau_0 \sim \mathcal{E}(10)$. This network prior enables to explore a large network space, while imposing more weights on networks with 1 or 2 reticulations (see Fig A of S1 Text). Recall that network A is a 1-reticulation network, whereas networks B and C are 2-reticulation networks. However, in order to limit the computational burden for network C (and for estimating continuous parameters on network A), we slightly modified the prior by bounding the number of reticulations to 2. Last, on network B, the analysis was performed by bounding the number of reticulations to 3 in order to compare SNAPPNET's results with those obtained by MCMC_BiMarkers [54]. We refer to Figs B and C in S1 Text for illustrations of the "bounded" prior.

**MCMC convergence.** To track the behaviour of the Bayesian algorithm, we used the Effective Sample Size (ESS) criterion [68]. We assumed that MCMC convergence was reached and that enough "independent" observations were sampled, when the ESS values for all model parameters are greater than 200 (see https://beast.community/ess_tutorial). This threshold is commonly adopted in the MCMC community. The first 10% samples were discarded as burn-in and the ESSs were computed on the remaining observations, using the Tracer software [69]. When we could not reach ESSs of 200, the ESS threshold is specified in the text. In the following, when speaking of a specific ESS value, we refer to the ESS computed for the posterior density function of the sampled networks (first value reported by Tracer). In order to estimate posterior distributions, we only sampled the MCMC every 1000 iterations. This was done to reduce autocorrelation across the sampled networks.

Note that here we do not attempt to measure an ESS of the network topologies sampled by the MCMC. While approaches to do this have been proposed for tree topologies [70], adapting such approaches to network topologies lies beyond the scope of this paper (see also the Discussion). Topological convergence was only assessed by inspecting the similarity between the results obtained for different MCMC replicates.

**Accuracy of SNAPPNET.** In order to evaluate SNAPPNET's ability to recover the true network topology, the posterior probability of the true topology was estimated by taking the proportion of sampled network topologies matching the true topology. Note that unlike previous works [54], we did not use a measure of topological dissimilarity, because most of the proposed measures can equal 0 even when the network topologies are different [38, 71]. In order to verify whether a sampled network and the true network have the same topologies, we used the isomorphism tester program available at https://github.com/igel-kun/phylo_tools. We report the average (estimated) posterior probability of the true network topology over the different replicates.

For some networks, we also investigated the ability to estimate continuous parameters, including network length (the sum of all branch lengths) and network height (the distance between the root and the leaves).

## Real data study on rice

In order to assess the performance of our method on real data, we addressed the case of rice, both a prominent crop and a well-studied advanced plant model for which extensive data is available. We used genomic data extracted from [72] and [73]. We focused on 24 representative varieties (see Table C and Fig M in S1 Text) spanning the four main rice subpopulations of cultivars (Indica, Japonica, cAus and cBasmati) as well as the different types of wild rice *O. rufipogon* that are suspected to have been involved in the origin of cultivated rice. We built three random data sets, keeping a large panel of Asian countries. Data set 1 contains only one variety per subpopulation, whereas data sets 2 and 3 contain two varieties per subpopulation

(cf. Tables D and E in S1 Text). For each of the 12 chromosomes, we sampled 1k SNPs having only homozygous alleles. Following recommendations of [19], the SNPs were chosen for each of the 12 chromosomes to be as separated as possible from one another to avoid linkage between loci, though [54] has shown this kind of analysis is quite robust to this bias. The concatenation of these SNPs lead to 12k whole-genome SNP data sets on the selected rice varieties.

SNAPPNET was run again discarding the first 10% of samples as burn-in and sampling the MCMC every 1000 iterations. The number of reticulations was bounded by two for data sets 1 and 3. On data set 2, in order to obtain results in a reasonable amount of time (cf. the Results section), only one reticulation was finally allowed.

## Results

### Simulations

First, we compare the performances of SNAPPNET and `MCMC_BiMarkers` on data simulated with networks A and B (cf. Fig 7), already studied in [54], and the more complex C network. Second, we compare the two tools in terms of CPU time and memory required to compute the likelihood of network C and its variants. This step is usually repeated million times in an MCMC analysis, and is therefore critical for its overall efficiency. Note that focusing on a single likelihood calculation allows us to exclude the effect of the prior on the overall efficiency of the MCMC, and to only test the computational efficiency of the new algorithm to compute the likelihood implemented in SNAPPNET.

**Study of networks A and B.**   *1) Ability to recover the network topology*

Table 1 reports on the ability of SNAPPNET to recover the correct topology of networks A and B. As in [54], we simulated one individual for each species. Note that under this setting, population sizes $\theta$ corresponding to external branches are unidentifiable, as there is no coalescence event occurring along these branches. We studied different densities of markers and different priors on $\theta$. Besides, we focused on either a) the true prior $\Gamma(1,200)$ with $\mathbb{E}(\theta) = 0.005$, b) the incorrect prior $\Gamma(1,1000)$ with $\mathbb{E}(\theta) = 0.001$, or c) the incorrect prior $\Gamma(1,2000)$ with $\mathbb{E}(\theta) = 5 \times 10^{-4}$. Last, in order to compare our results with [54], we considered $u$ and $v$, the mutation rates, as known parameters. Indeed, `MCMC_BiMarkers` relies on the operators of [52] that do not allow changes of these rates.

First consider simulations under the true prior. As shown in Table 1, in presence of a large number of markers, SNAPPNET recovered networks A and B with high posterior probability. In particular, when $m = 100,000$ sites were used, the posterior distributions were only concentrated on the true networks. For $m = 10,000$, the average posterior probability of network A is

**Table 1. Average posterior probability of the correct topology (for networks A and B, see Fig 7) obtained by running SNAPPNET on simulated data.** Results are given as a function of the number of sites and as a function of the hyperparameter values $\alpha$ and $\beta$ for the prior on $\theta$ ($\theta \sim \Gamma(\alpha, \beta)$ and $\mathbb{E}(\theta) = \frac{\alpha}{\beta}$). Here, one lineage was simulated per species. Constant sites are included in the analysis, the rates $u$ and $v$ are considered as known, and 20 replicates are considered for each simulation set up (criterion ESS > 200; $d \sim \mathcal{E}(0.1)$, $r \sim \text{Beta}(1, 1)$, $\tau_0 \sim \mathcal{E}(10)$ for the network prior).

| | Network A | | | Network B | | |
|---|---|---|---|---|---|---|
| **Number of sites**<br>**Hyperparameters** | **1,000** | **10,000** | **100,000** | **1,000** | **10,000** | **100,000** |
| **True** ($\alpha = 1, \beta = 200, \frac{\alpha}{\beta} = 0.005$) | 0% | 100% | 100% | 0% | 81.25% | 100% |
| **Incorrect** ($\alpha = 1, \beta = 1000, \frac{\alpha}{\beta} = 0.001$) | 0% | 94.73% | 91.30% | 0% | 80% | 95.65% |
| **Incorrect** ($\alpha = 1, \beta = 2000, \frac{\alpha}{\beta} = 5 \times 10^{-4}$) | 0% | 100% | 80% | 0% | 85% | 85.71% |

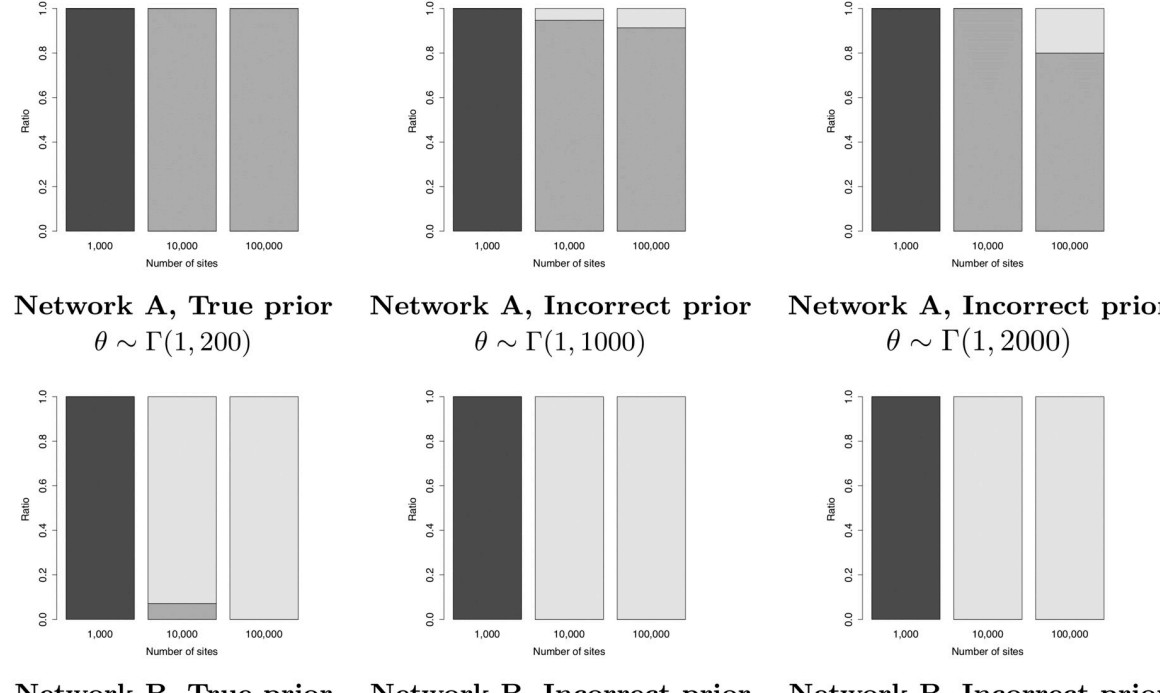

**Fig 9. The ratio of trees (black), 1-reticulation networks (dark grey), 2-reticulations networks (light gray), sampled by SNAPPNET, under the different simulations settings studied in Table 1.** Recall that networks A and B contain 1 and 2 reticulations, respectively.

again 100%, whereas that of B is lower (81.25%). This is not surprising since network B is more complex than network A. Our results are consistent with those of [54], who found that `MCMC_BiMarkers` required 10,000 sites to infer precisely networks A and B. (Recall that we did not rerun `MCMC_BiMarkers` on data simulated from networks A and B).

However, for a small number of sites ($m = 1,000$), we observed differences between SNAPPNET and `MCMC_BiMarkers`: SNAPPNET always inferred trees (see Fig 9), whereas `MCMC_BiMarkers` inferred networks. For instance, on Network A, `MCMC_BiMarkers` inferred a network in approximately 75% of cases, whereas SNAPPNET supported the tree ((((Q,A),L),R),C) with average posterior probability 78.71%. Interestingly, this tree can be obtained from network A by removing the hybridization branch with smallest inheritance probability. Details on the trees inferred by SNAPPNET for this setting are given in Table A of S1 Text.

Similarly, on network B that hosts 2 reticulations, for $m = 1,000$ `MCMC_BiMarkers` almost always inferred a 1-reticulation network [54], whereas SNAPPNET hesitated mainly between two trees, (((Q,R),L),(A,C)) and (((Q,L),R),(A,C)), with average posterior probabilities 35.28% and 28.54%, respectively. This different behavior among the two tools is most likely due to the fact that their prior models differ. With only 1,000 markers, `MCMC_BiMarkers` and SNAPPNET were both unable to recover network B.

Now consider simulations based on incorrect priors. This mimics real cases where there is no or little information on the network underlying the data. Recall that these priors are incorrect since $\mathbb{E}(\theta)$ is either fixed to 0.001 or $5 \times 10^{-4}$, instead of being equal to the true value 0.005. In other words, these priors underestimate the number of ILS events in the data. When

considering as few as 1,000 sites, SNAPPNET only inferred trees (cf. Table A in S1 Text), whereas MCMC_BiMarkers mostly inferred networks [54]. For $m$ = 10,000 and $m$ = 100,000 sites, SNAPPNET inferred network A with high posterior probability. In the rare cases where the true network was not sampled, SNAPPNET inferred a network with two reticulations (see Fig 9). The bias induced by incorrect priors (underestimating ILS) led the method to fit the data by adding supplementary edges to the network. On network B, SNAPPNET's posterior distribution remained concentrated on the correct topology, and interestingly, for $m$ = 10,000 and $m$ = 100,000 sites, SNAPPNET sampled exclusively 2-reticulation networks (see Fig 9). To sum up, SNAPPNET's ability to recover the correct network topology did not really deteriorate with incorrect priors.

*2) Ability to estimate continuous parameters for network A*

Recall that in our modelling, the continuous parameters are branch lengths, inheritance probabilities $\gamma$, population sizes $\theta$ and instantaneous rates ($u$ and $v$). As in [53], we also studied the network length and the network height, that is the sum of the branch lengths and the distance between the root node and the leaves, respectively. In order to evaluate SNAPPNET's ability to estimate continuous parameters, we will focus here exclusively on network A (following [54]). Analogous results for networks B and C can be found in Figs D-G in S1 Text.

For network A, we considered the case of two lineages in each species. Indeed, under this setting, $\theta$ values are now identifiable for external branches: the expected coalescent time is here $\theta/2$, that is to say $2.5 \times 10^{-3}$, which is a smaller value than all external branch lengths. In other words, a few coalescent events should happen along external branches. For these analyses, we considered exclusively the true prior on $\theta$ and we bounded the number of reticulations to 2 (as in [54]) in order to limit the computational burden. In the following, we consider the cases where a) input markers can be invariant or polymorphic, and b) only polymorphic sites are considered.

*2a) Constant sites included in the analysis*

Before describing results on continuous parameters, let us first mention results regarding the topology. Although the number of lineages was increased in comparison with the previous experiment, SNAPPNET still sampled exclusively trees for $m$ = 1,000, and always recovered the correct topology for $m$ = 10,000 and $m$ = 100,000. Note that for $m$ = 1,000, we observed that generated data sets contained 78% invariant sites on average given the parameters of the simulation, so that such simulated data sets only contained on average 220 variable sites.

In order to limit the computational burden, the analysis for $m$ = 100,000 relied only on 17 replicates with ESS > 200. Fig 10 reports on the estimated network height and the estimated network length. As expected, the accuracy increased with the number of sites. Fig 11 shows the same behaviour, regarding the inheritance probability $\gamma$, the rates $u$ and $v$. Fig 12 is complementary to Fig 10, since it reports on the estimated node heights. All node heights were estimated quite accurately, which is not surprising in view of the results on the network length. Fig 13 is dedicated to population sizes. For external branches, SNAPPNET's was able to estimate $\theta$ values very precisely. Performances slightly deteriorated on internal branches (see the box plots, from number 6 to number 12) whose $\theta$ values were underestimated (see the medians) and showed a higher posterior variance. This phenomenon was also observed for MCMC_BiMarkers [54, Fig 7 obtained under a different setting].

*2b) Only polymorphic sites included in the analysis*

In order to control for the fact that this analysis relies only on polymorphic sites, the likelihood of the data for a network $\Psi$ becomes a conditional likelihood equal to $\mathbb{P}(X_1, \ldots, X_m \mid \Psi) / \mathbb{P}(\text{"the } m \text{ sites are polymorphic"}|\Psi)$, due to Bayes' rule.

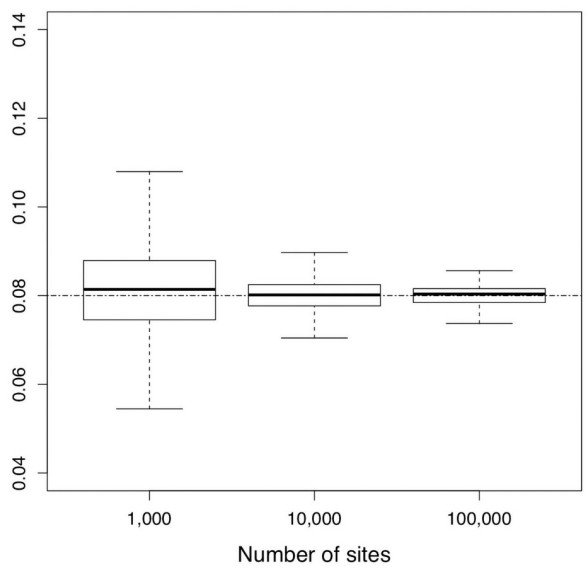
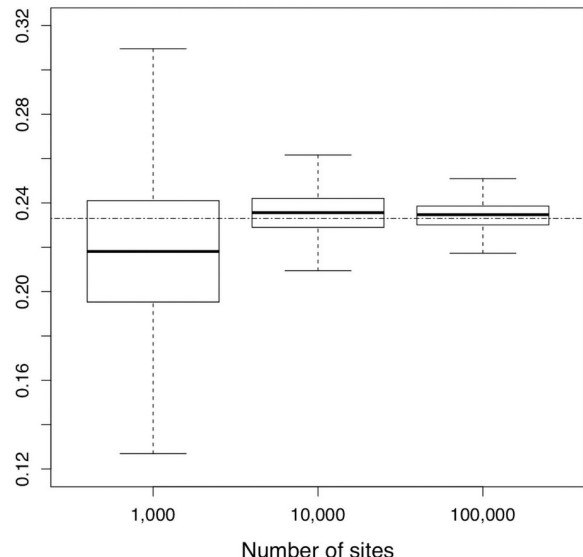

**Network height**                                                    **Network length**

**Fig 10. Estimated height and length for network A (see Fig 7), as a function of the number of sites.** Heights and lengths are measured in units of expected number of mutations per site. True values are given by the dashed horizontal lines. Two lineages per species were simulated. Constant sites are included in the analysis, and 20 replicates are considered for each simulation set up (criterion ESS > 200; $\theta \sim \Gamma(1, 200)$, $d \sim \mathcal{E}(0.1)$, $r \sim$ Beta(1, 1), $\tau_0 \sim \mathcal{E}(10)$ for the priors, number of reticulations bounded by 2 when exploring the network space).

Before focusing on continuous parameters, let us describe results regarding the topology. As mentioned in [54], polymorphic sites are considered as most informative to recover the topology. For $m = 1{,}000$, SNAPPNET now recovers the correct topology of network A with high frequency in 94.45% of samples. SNAPPNET always sampled the true network for $m = 10{,}000$ and $m = 100{,}000$. In order to reduce the computational burden for $m = 100{,}000$, our analysis relied on the 12 replicates that achieved ESS > 100.

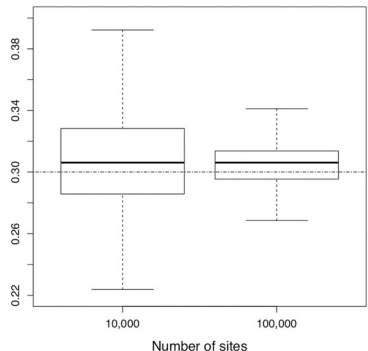
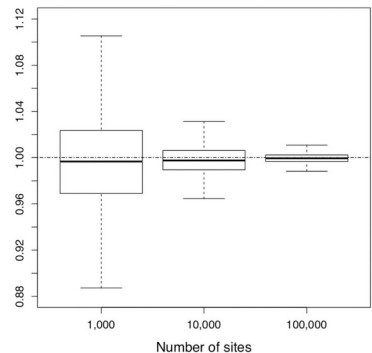
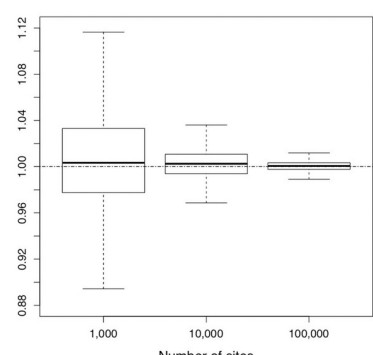

**Inheritance probability $\gamma$**     **Instantaneous rate $u$**     **Instantaneous rate $v$**

**Fig 11. Estimated inheritance probability and instantaneous rates for network A (see Fig 7), as a function of the number of sites. True values are given by the dashed horizontal lines.** Same framework as in Fig 10.

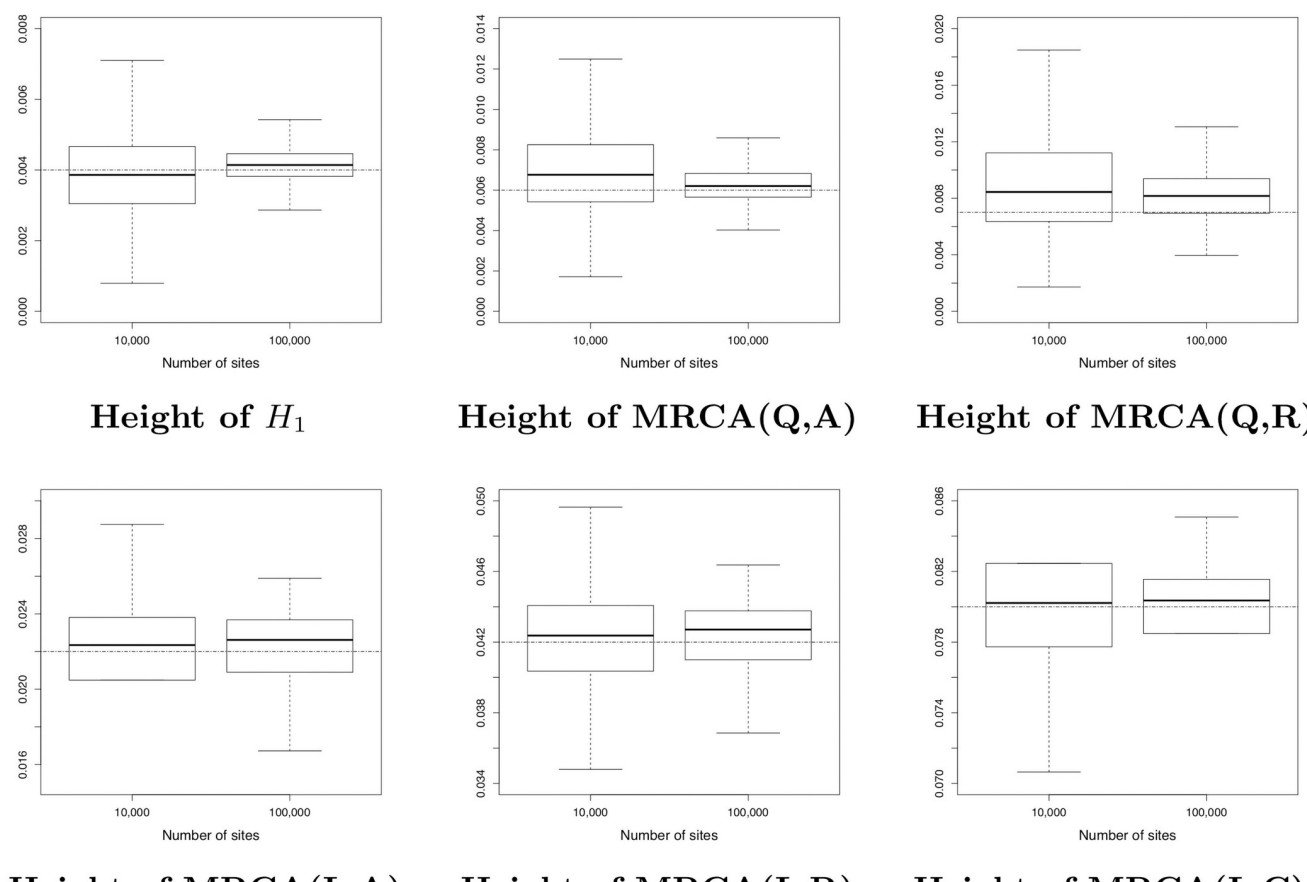

**Fig 12. Estimated node heights of network A (see Fig 7), as a function of the number of sites.** Heights are measured in units of expected number of mutations per site. True values are given by the dashed horizontal lines. Same framework as in Fig 10. The initials MRCA stand for "Most Recent Common Ancestor".

Next, the same analysis was performed without applying the correction factor $\mathbb{P}(\text{"the m sites are polymorphic"}|\Psi)$, which is done by toggling an option within the software. For $m = 1,000$, the average posterior probability of network A dropped to 23.81%, while for $m = 10,000$ and $m = 100,000$, it remained relatively high (i.e., 95.24% and 95.65%, respectively). Using the correct likelihood computation is important here.

We also highlight that for $m = 100,000$, the sampler efficiency (i.e. the ratio ESS/nb iterations without burn-in) was much larger when the additional term was omitted ($1.75 \times 10^{-4}$ vs. $2.55 \times 10^{-5}$). It enabled us to consider 20 replicates with ESS > 200 in this new experiment.

Let us move on to the estimation of continuous parameters. Figs H-K in S1 Text illustrate results obtained from the experiment incorporating the correction factor. As previously, the network height, the network length, the rates $u$ and $v$, the inheritance probability $\gamma$ and the node heights were estimated very precisely. As expected, the accuracy increased with the number of sites. Estimated $\theta$ values were very satisfactory for external branches, whereas a slight bias was still introduced on internal branches. Last, for the analysis without the correction factor, we observed a huge bias regarding network height and network length (cf Fig L in S1 Text). Surprisingly, the rates $u$ and $v$ were still very accurately estimated.

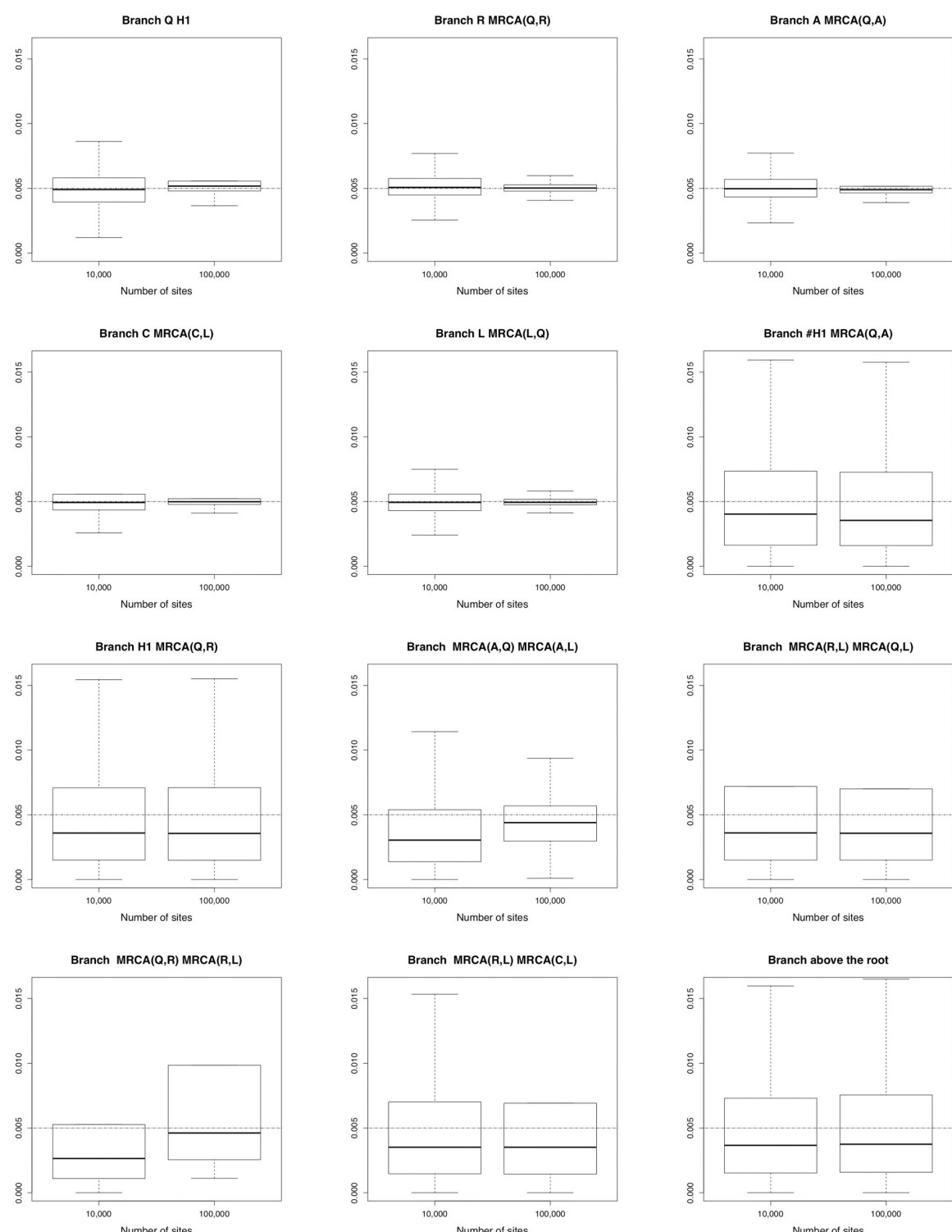

**Fig 13. Estimated population sizes θ for each branch of network A (see Fig 7), as a function of the number of sites.** True values are given by the dashed horizontal lines. Same framework as in Fig 10. The initials MRCA stand for "Most Recent Common Ancestor".

**Table 2. Average posterior probability (PP) of the topology of network C obtained by running SNAPPNET on data simulated from network C.** Results are given as a function of the number of sites and as a function of the number of lineages sampled in hybrid species B and C (either both 1 or both 4). Only one lineage was sampled in every other species. Constant sites are included in the analysis and the rates $u$ and $v$ are considered as known. Posterior probabilities are computed on the basis of replicates for which the criterion ESS>100 is fulfilled. The sampler efficiency (SE) is also indicated (true hyperparameter values for the prior on $\theta$, i.e. $\theta \sim \Gamma(1, 200)$; as a network prior $d \sim \mathcal{E}(0.1)$, $r \sim \mathrm{Beta}(1, 1)$, $\tau_0 \sim \mathcal{E}(10)$; number of reticulations bounded by 2 when exploring the network space).

| Number of lineages for B and for C | | Number of sites | | |
|---|---|---|---|---|
| | | **1,000** | **10,000** | **100,000** |
| 1 | PP | 0% (20 replicates) | 7.87% (20 replicates) | 54.9% (20 replicates) |
| | SE | $3.18 \times 10^{-4}$ | $3.47 \times 10^{-4}$ | $4.84 \times 10^{-3}$ |
| 4 | PP | 0% (20 replicates) | 50.00% (18 replicates) | 49.6% (8 replicates) |
| | SE | $7.63 \times 10^{-3}$ | $3.89 \times 10^{-4}$ | $2.65 \times 10^{-4}$ |

**Study of network C and its variants.** We focus here on network C (Fig 7) and its variants (Fig 8).

*1) Ability to recover the network topology*

Tables 2 and 3 report the ability of SNAPPNET and `MCMC_BiMarkers`, respectively, to recover the correct topology of network C. We considered one lineage in species O, A and D, and let the number of lineages in species B and C vary. We studied either a) 1 lineage, or b) 4 lineages, in these hybrid species. In order to limit the computational burden for SNAPPNET, the ESS criterion was decreased to 100 and the number of reticulations was also bounded by 2.

In order to closely mimic what was done in [54] for networks A and B, we let `MCMC_BiMarkers` run for 1,500,000 iterations instead of adopting an ESS criterion. Data were simulated with `simBiMarker` [54]. Indeed, like SIMSNAPP, SIMSNAPPNET generates only count data (the number of alleles per site and per species). In contrast, `simBiMarker` generates actual sequences, a prerequisite for running `MCMC_BiMarkers`. The commands used under the 4 lineages scenario are given in Section 5 of S1 Text. Note that, to calibrate the network prior of `MCMC_BiMarkers`, the maximum number of reticulations was set to 2, and the prior Poisson distribution on the number of reticulation nodes was centered on 2.

As expected, SNAPPNET's ability to recover the correct network topology increased with the number of sites and with the number of lineages in the hybrid species (see Table 2). For instance, in the presence of one lineage in hybrid species B and C, the posterior probability of network C increased from 7.87% for $m = 10,000$ to 54.90% for $m = 100,000$. In the same way, when 4 lineages were considered instead of a single lineage, we observed an increase from 7.87% to 50.00% for $m = 10,000$. Note that the average posterior probability of 49.60% reported for $m = 100,000$ and 4 lineages, is based only on 8 replicates.

**Table 3. Average posterior probability (PP) of the topology of network C obtained by running `MCMC_BiMarkers` on data simulated from network C.** Results are given as a function of the number of sites and as a function of the number of lineages sampled in hybrid species B and C (either both 1 or both 4). Only one lineage was sampled in every other species, constant sites are included in the analysis, and the rates $u$ and $v$ are considered as known. $1.5 \times 10^6$ iterations are considered. $\overline{\mathrm{ESS}}$ is the average ESS over the different replicates, and SE stands for the sampler efficiency.

| Number of lineages for B and for C | | Number of sites | | |
|---|---|---|---|---|
| | | 1,000 | 10,000 | 100,000 |
| 1 | PP | 0% (20 replicates) | 4.84% (20 replicates) | 0% (20 replicates) |
| | SE | $9.70 \times 10^{-5}$ | $3.10 \times 10^{-5}$ | $3.60 \times 10^{-5}$ |
| | $\overline{\mathrm{ESS}}$ | 126.08 | 40.38 | 46.80 |
| 4 | PP | 0% (20 replicates) | 0% (12 replicates) | 0% (9 replicates) |
| | SE | $2.38 \times 10^{-4}$ | $8.53 \times 10^{-5}$ | $1.03 \times 10^{-5}$ |
| | $\overline{\mathrm{ESS}}$ | 309.00 | 110.90 | 159.96 |

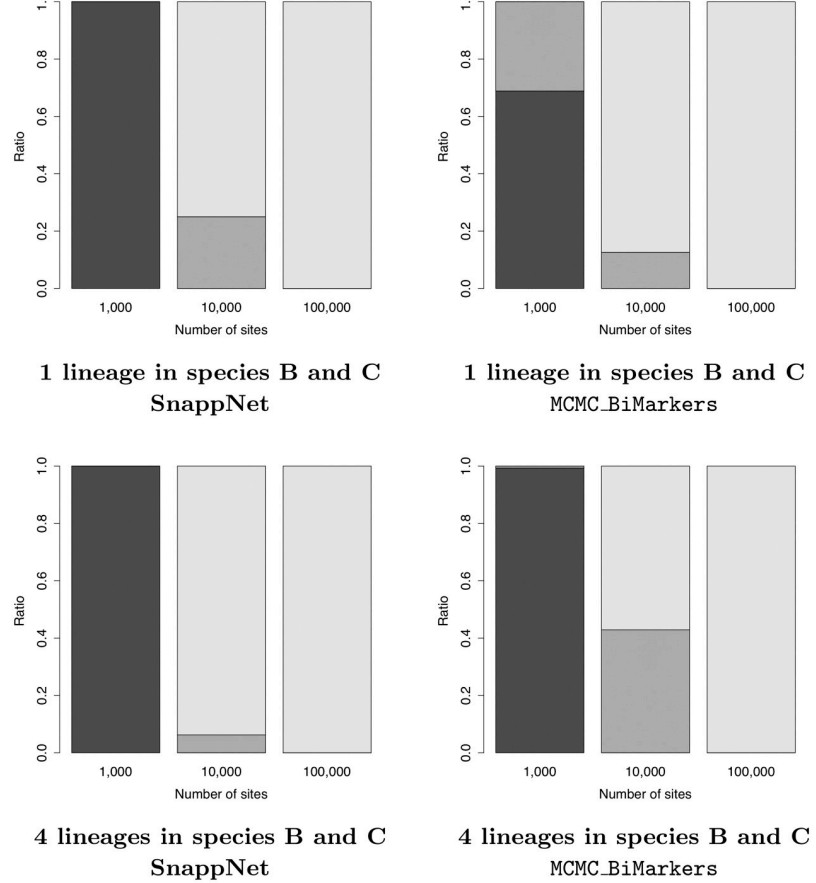

**Fig 14. Frequency of trees (black), 1-reticulation networks (dark grey), 2-reticulations networks (light gray) sampled by SNAPPNET and `MCMC_BiMarkers`, when data were simulated from Network C (see Tables 2 and 3).** Recall that network C contains 2 reticulations.

Surprisingly, in most cases studied, `MCMC_BiMarkers` was unable to recover the true topology of network C. The different behaviors of `MCMC_BiMarkers` and SNAPPNET may be due to the different network priors. Indeed, while the frequency of trees, 1-reticulation networks and 2-reticulations networks sampled by the two methods were globally similar (cf. Fig 14), we remarked that `MCMC_BiMarkers` seems to be unable, for these data sets, to sample networks with two reticulations on top of each other. Alternatively, we may be in the presence of failed or partial convergence of the MCMC process. Note the small ESS values for `MCMC_BiMarkers`, especially when only one lineage was sampled in hybrid species B and C. However, we attempted increasing the number of iterations from $1.5 \times 10^6$ to $12 \times 10^6$ and `MCMC_BiMarkers` was still unable to recover network C, despite larger ESS values (see Table B in S1 Text). We note here that SNAPPNET was ran for a maximum 804,000 iterations for 10,000 sites, and a maximum of 555,000 iterations for 100,000 sites.

*2) CPU time and required memory*

To compare the CPU time and memory required by SNAPPNET and `MCMC_BiMarkers` on a single likelihood calculation, we focused on network C (see Fig 7), with and without outgroup (i.e. the species O), and networks C(3) and C(4), again with and without outgroup (see Fig 8). The simulations protocol used here is similar to that used in the previous sections, where here we fixed 10 lineages in species C and one lineage in the other species, $m = 1,000$

**Table 4. Computational efficiency of calculating a single likelihood value in SNAPPNET and `MCMC_BiMarkers` for networks C, C(3) and C(4).** 10 lineages are sampled in species C and 1 lineage in other species. Average and standard deviation are reported.

| | CPU time | | Memory | |
|---|---|---|---|---|
| | SNAPPNET (in minutes) | `MCMC_BiMarkers` (in hours) | SNAPPNET (max in GB) | `MCMC_BiMarkers` (max in GB) |
| Network C without outgroup | 2.62 ± 0.04 | 14.58 ± 0.50 | 1.67 ± 0.03 | 8.76 ± 0.02 |
| Network C | 5.63 ± 0.16 | 33.46 ± 1.31 | 2.00 ± 0.09 | 8.79 ± 0.02 |
| Network C(3) without outgroup | 14.21 ± 0.56 | ? | 2.19 ± 0.01 | < 64 |
| Network C(3) | 24.69 ± 0.64 | ? | 2.21 ± 0.06 | <64 |
| Network C(4) without outgroup | 45.47 ± 1.44 | ? | 2.63 ± 0.60 | > 128 |
| Network C(4) | 70.98 ± 3,16 | ? | 3.17 ± 0.81 | > 128 |

sites and 20 replicates per each network. The likelihood calculations were run on the true network.

The experiments were executed on a full quad socket machine with a total of 512GB of RAM ($4 * 2.3$ GHz AMD Opteron 6376 with 16 Cores, each with a RDIMM 32Go Quad Rank LV 1333MHz processor). The jobs that did not finish within two weeks, or required more than 128 GB, were discarded.

The results are reported in Table 4. SNAPPNET managed to run for all the scenarios within the two weeks limit: on average within 2.62 minutes and using 1.67 GB on network C without outgroup, within 5.63 minutes and using 2 GB on network C with outgroup, within 14.21 minutes and using 2.19 GB on network C(3) without outgroup, within 24.69 minutes and using 2.21 GB on network C(3) with outgroup, within 45.47 minutes and using 2.63 GB on network C(4) without outgroup, and finally, within 70.98 minutes and using 3.17 GB on network C(4) with outgroup.

We were able to run `MCMC_BiMarkers` for all replicates of the network C, and we can thus compare its performance with that of SNAPPNET. From Table 4, we see that SNAPPNET is remarkably faster that `MCMC_BiMarkers`, needing on average only 0.29% of the time and 21% of the memory required by `MCMC_BiMarkers`. `MCMC_BiMarkers` needed more than 2 weeks for all scenarios on the C(3) network (requiring less than 64 GB), thus no run time is available for these scenarios. The same holds for the C(4) network scenarios, but for a different reason: all runs needed more than 128 GB each, and were discarded.

In Section 8 of S1 Text we provide the results of additional experiments on simulated data. In Section 8.1, we assess whether SNAPPNET's MCMC sampler can adequately sample from network space. In Section 8.2 we assess how population size priors and network priors influence SNAPPNET's inferences.

## Real data analysis

Real data derived from recent studies on rice were used to illustrate the application of SNAPPNET.

Diversity among Asian rice cultivars is structured around two major types which display worldwide distributions, namely Japonica and Indica, and two types localised around the Himalayas, namely *circum* Aus (cAus) and *circum* Basmati (cBasmati) [73, 74]. Japonica and Indica each have several subgroups with geographical contrast (see [73] as the most detailed description). Domestication scenarios that have been put forwards since the availability of whole genome sequences propose one to three domestications corresponding either to an early pivotal process in Japonica [72], or to multiple parallel dynamics in Japonica, Indica and cAus [12, 27], depending on whether they consider the contribution of domestication alleles

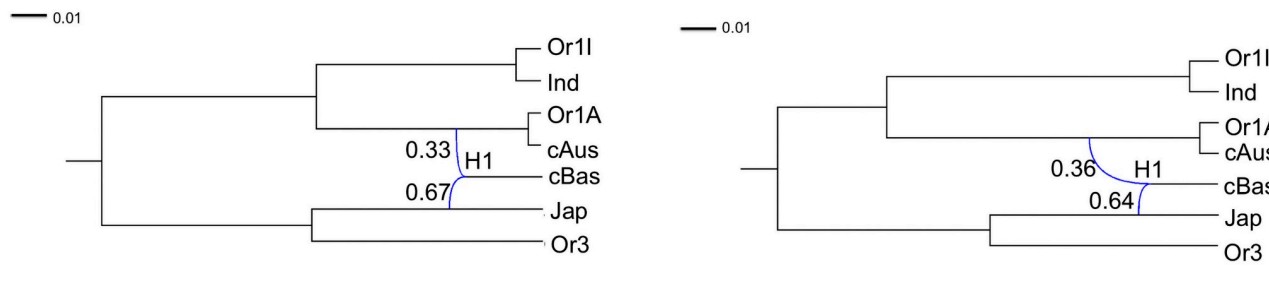

**Fig 15. The two networks obtained for data set 1 with only one variety per subpopulation.** Each network corresponds to the posterior mean of the distribution sampled by SNAPPNET. Inheritance probabilities are reported above reticulation edges and branch lengths are given in units of expected number of mutations per site (see the scale at the top left).

by the Japonica origin as predominant or as one among others. cBasmati has been posited as a specific lineage within Japonica [72] or as a secondary derivative from admixture between Japonica and a local wild rice close to cAus [75], or between Japonica and cAus with the contribution of one or several additional cryptic sources [76].

The most advanced studies of wild rice [72] recognize three populations designated Or-I to Or-III (Or for *Oryza rufipogon*), of which Or-I and Or-III are closely related to cultivars and Or-II is not. Using a data set constructed in [73], we compared wild rices to cultivars on the basis of ca. 2.5 million SNPs (cf. Fig M in S1 Text) and we selected representatives of Japonica, Indica, cAus and cBasmati as well as wild rices Or-III, closer to Japonica and cBasmati, and Or-I, closer either to Indica (Or-Ii) or to cAus (Or-Ia). For clarity in our subsequent use, we call the wild forms Or3, Or1I and Or1A, respectively. We made data sets of different sample sizes, including either one or two varieties per subpopulation. The studied subpopulations are the 4 groups of cultivars (Japonica, Indica, cAus, cBasmati), and different types of wild rice (Or3, Or1A, Or1I), consistent with the classification by [72]. The 3 data sets we constructed are described in the Materials and methods.

In Fig 15, we report results for data set 1, which includes only one variety per subpopulation (cf. Table D in S1 Text). We studied two different samplings of 12k SNPs along the whole genome alignment. For each sampling, we ran two independent Markov chains with different starting points, for 10 million iterations. To assess the convergence of SNAPPNET on data set 1, (a) the ESS of the posterior distribution was checked for each chain, (b) the trace plots of the different parameters and their associated ESS were examined and (c) the two posterior distributions corresponding to the two independent chains were compared (see Fig N and Table F in S1 Text). In view of these results, SNAPPNET reached stationarity. The ESS of the posterior distribution took the values 844 (resp. 971), 1159 (resp. 535) for the two different chains of the first (resp. second) sampling. All the networks sampled by the MCMC had the same topology with one reticulation only. For both genome samplings the lineages associate Or1I with Ind, Or1A with cAus and Or3 with Jap, respectively, while the reticulation conjugates Jap with (Or1A/cAus), the common precursor of Or1-A/cAus, with a dosage ratio close to 2:1, to yield cBas.

Next, we tackled a larger data set, data set 2, containing two varieties per subpopulation (see Table E in S1 Text). Two different chains corresponding to two different samplings of 12k SNPs along the whole genome alignment were run. The number of reticulations was bounded by one in order to reach convergence in a reasonable amount of time: after three months and

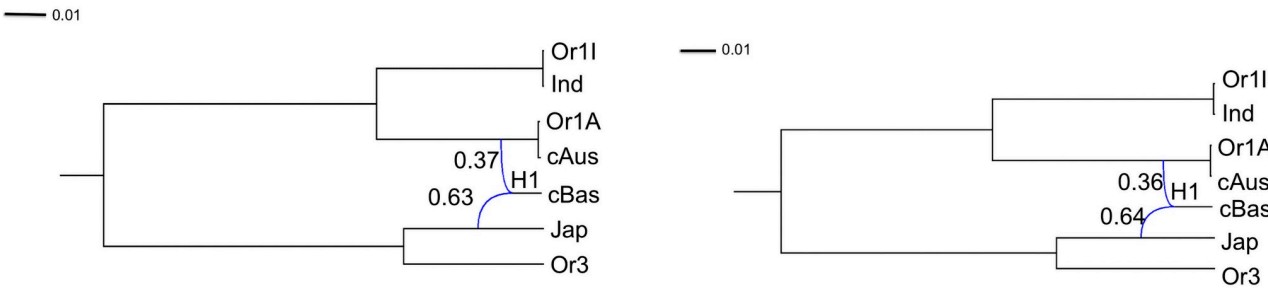

**Fig 16. The two networks obtained for data set 2 with two varieties per subpopulation.** Each network corresponds to the posterior mean of the distribution sampled by SNAPPNET. Inheritance probabilities are reported above reticulation edges and branch lengths are given in units of expected number of mutations per site (see the scale at the top left).

half of computations, the ESS took the values 227 and 201 for the first and the second chain, respectively. Fig 16 illustrates the two networks obtained for the two different samplings. Each network corresponds to the posterior mean of the sampled distribution. Note that in both cases, the posterior distribution was concentrated on a single topology. The two genome samplings yield networks very similar to one another and remarkably close to that revealed with data set 1. The reticulation that was allowed again conjugates the Jap lineage with the common precursor of subpopulations Or1A and cAus. In contrast, after 6 months of calculations, SNAPP-NET had still not reached the stationary regime for the two different samplings, when a maximum of 2 reticulations was imposed.

We also investigated another data set, data set 3, including two varieties per subpopulation and 12k SNPs for a different taxon sampling (see Table E in S1 Text). In this case, large ESS values were observed when SNAPPNET was allowed to infer networks with 2 reticulations: the ESS was estimated at 373 after having let SNAPPNET run for 7 months. The maximum a posteriori (MAP) network is represented in Fig 17. For this data set, the resulting topology again features a single reticulation, although two were allowed. It also conjugates Jap with the precursor (Or1A/cAus) of Or1A and cAus to produce cBas. Yet the composition is more unbalanced towards Jap (0.85) and (Or1A/cAus) appears involved very close to the Or1A vs cAus initial divergence. Given this proximity, it was useful to describe the three networks retained by SNAPPNET during the MCMC process (Fig 18). The first one (67%) features a conjugation

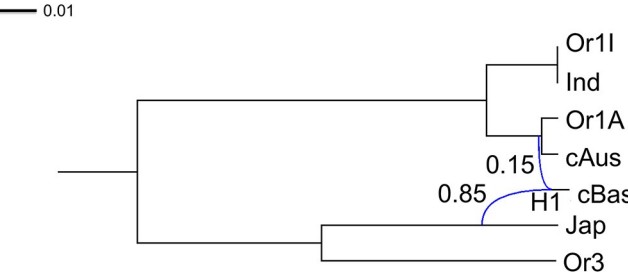

**Fig 17. The MAP phylogenetic network obtained for data set 3 with two varieties per subpopulation.** Inheritance probabilities are reported above reticulation edges and branch lengths are given in units of expected number of mutations per site (see the scale at the top left).

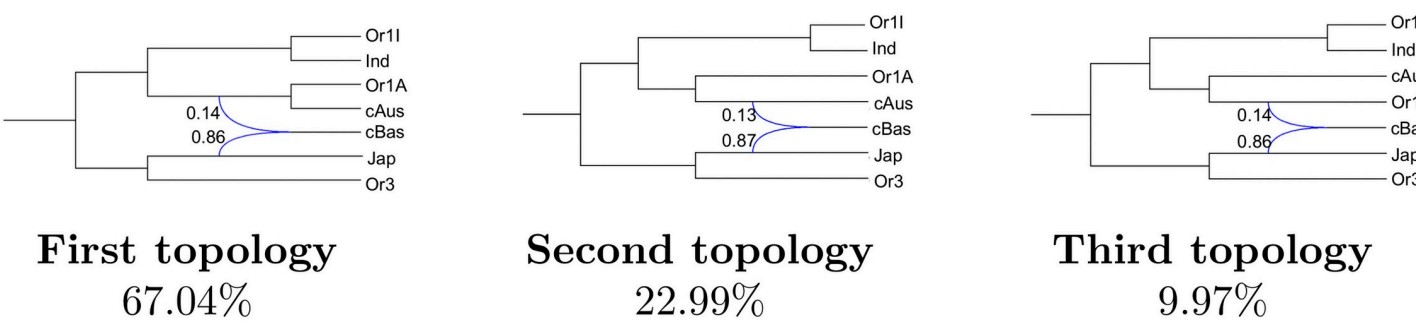

**Fig 18. The three topologies sampled by SNAPPNET when data set 3 was considered.** Reported inheritance probabilities for each topology are averages on sampled observations.

between Jap and (Or1A/cAus), while the second one (23%) conjugates Jap with cAus and the third one (10%) conjugates Jap with Or1A in the origin of cBas.

Altogether the various networks inferred by SNAPPNET reveal stable features:

- the correspondence between wild subpopulations and cultivated subpopulations which point at three pillars for rice, namely Japonica, Indica and cAus

- the early divergence of Japonica, that predates the one between Indica and cAus

- the earlier divergence between wild and cultivated forms within the Japonica pillar

- the mobilisation of early Japonica cultivars to combine with the cAus pillar to produce the fourth varietal type cBas

- the indication that this hybridization may have occured before the domestication of cAus.

The latter item yet displays uneven strength levels between the various data sets. The first four items confirm the latest interpretations of massive analyses among rice specialists. Wild rice displays broad diversity and some of the wild subpopulations have been specifically involved in the emergence of cultivated forms. While the most ancient domestication occurred in China to produce Japonica cultivars, two other important foundations, namely Indica and cAus, contributed to the diversity of current rice cultivars. Early hybridization between Japonica cultivars and an ancestor, presumably wild, of current cAus cultivars and related wild forms resulted in the evolution of cBasmati cultivars.

## Discussion

In this paper, we introduced a new Bayesian method, SNAPPNET, dedicated to phylogenetic network inference. SNAPPNET has similar goals as `MCMC_BiMarkers`, a method recently proposed by Zhu et al. [54], but differs from this method in two main aspects. The first difference is due to the way the two methods handle the complexity of the sampled networks. Unlike binary trees that have a fixed number of branches given the number of considered species, network topologies can be of arbitrary complexity. Their complexity directly depends on the number of reticulations they contain. In MCMC processes, the complexity of sampled networks is regulated by the prior. `MCMC_BiMarkers` uses descriptive priors: more precisely, it assumes a Poisson distribution for the number of reticulation nodes and an exponential distribution for the *diameter* of reticulation nodes [51, 52, 54]. In contrast, SNAPPNET's prior is based on that of Zhang et al., which explicitly relies on speciation and hybridization rates and is extendable to account for extinction and incomplete sampling [53].

Our simulation study may provide some insight on the influence of these different priors. On two networks of moderate complexity (networks A and B), SNAPPNET and `MCMC_BiMarkers` presented globally similar results. Indeed, when we considered numbers of sites that are largely achieved in current phylogenomic studies (i.e. 10,000 or 100,000 sites), both methods were able to recover the true networks under this realistic framework. However, in presence of only a few sites (1,000 sites) which is unusual nowadays but still can be the case for poorly sequenced organisms, `MCMC_BiMarkers` recovered the correct topology with higher posterior probability than SNAPPNET. On the other hand, when focusing on a more complex network (network C) containing reticulation nodes on top of one another, the converse appeared to be true. With sufficiently large datasets, SNAPPNET recovered the correct scenario in approximately 50% of samples whereas `MCMC_BiMarkers` inferred this history in less than 5% of cases. Although these differences may be due to the different network priors used by the two methods, more work is needed to elucidate the reasons behind them. To conclude the discussion on priors, we also observed that, on simulated data, SNAPPNET's accuracy did not really deteriorate with incorrect priors on population sizes, although assuming a prior distribution skewed towards small population sizes has a tendency to favor hybridization over ILS as an explanation for non tree-like signals. Similar robustness properties were observed by [54] for `MCMC_BiMarkers`.

The second major difference between `MCMC_BiMarkers` and SNAPPNET lies in the way they compute the likelihood of a network. This step is at the core of the Bayesian analysis. According to the authors of `MCMC_BiMarkers`, this remains a major computational bottleneck and limits the applicability of their methods [59]. To understand the origin of this bottleneck, recall that the MCMC process of a Bayesian sampling explores a huge network space and that, at each exploration step, computing the likelihood is by far the most time consuming operation. Moreover, we need sometimes millions of runs before the chain converges. Thus, likelihood computation is a key factor on which to operate to be able to process large data sets.

The likelihood computation of `MCMC_BiMarkers` consists in a bottom-up traversal, from the leaves to the root. Each time a reticulation node $r$ is visited, the partial likelihoods must be decomposed following all the possible ways the lineages reaching $r$ can be assigned to the two parent populations of $r$. These partial likelihoods will be merged back only when the traversal reaches a lowest articulation node [54], or in other words the root of the blob to which $r$ belongs (a *blob* is a maximal biconnected subgraph [65], see also S1 Text). For every other reticulation $r'$ reached before the root of the blob, the decomposition above is applied again. As a result, the time required to process a blob grows exponentially with the number of reticulations it contains. More precisely, the time complexity of the likelihood computation in `MCMC_BiMarkers` is in $O(sn^{4\ell+4})$, where $\ell$ is the *level* of the network and $s$ is the size of the species network.

Similarly to `MCMC_BiMarkers`, we compute the likelihood in a bottom-up traversal and when reaching a reticulation node $r$, we also take into account the various ways lineages could have split. But the originality of SNAPPNET is to compute *joint conditional probabilities* for branches above a same reticulation node $r$ (see the Materials and methods). The set of branches jointly considered increases when crossing other reticulation nodes in a same blob, but it can also decrease when crossing tree-nodes in the blob (i.e. nodes having one ancestor and several children). Of course, the time to compute each partial likelihood increases in proportion with the number of branches considered together. More precisely, SNAPPNET runs in $O(sn^{2\overline{K}+2})$, where $\overline{K}$ is the maximum number of branches simultaneously considered in a partial likelihood. The interest in depending on $\overline{K}$ instead of $\ell$ (the number of reticulations in a blob), is that for some blobs, we can resort to a bottom-up traversal of the blob that limits $\overline{K}$ to

a small constant and process the blob in polynomial time in $n$, while `MCMC_BiMarkers` still requires an exponential time in $\ell$.

Our results from simulated data confirm the above theoretical discussion. For a single likelihood evaluation, SNAPPNET was found to be orders of magnitude faster than `MCMC_BiMarkers` on networks containing reticulation nodes on top of one another. Besides, SNAPPNET required substantially less memory than `MCMC_BiMarkers`. These gains enable us to consider complex evolution scenarios in our Bayesian analyses.

In practice, SNAPPNET is a very useful tool for analyzing complex genomic data, as evidenced by our study about rice. Indeed, the most recent extensive genetic studies on this crop confirm and document the extent of genetic exchanges in various directions. Yet the same species consistently displays the reality of a simple classification scheme with only a few predominant types. Thus rice appears as a chance and a challenge for testing methods aiming to tackle phylogenetic resolution within a hybrid swarm. The application of SNAPPNET proves very efficient in resolving the three main phylogenetic pillars of current diversity in Asian rice [12, 77] and revealing a hybrid origin for the iconic varietal group cBasmati [75, 76]. The various data sets treated here suggest a contribution of Japonica cultivars at a high level, between 0.6 and 0.85. This rather broad range is not surprising given that this hybrid origin probably reflects numerous recent individual stories for very specific varieties rather than an old common story for a homogeneous lineage. On the other side, the second component of cBasmati derived from local sources in the North of the Indian subcontinent seems to date from before the evolution of cAus varieties. Here again, it is likely that many diverse events occurred resulting in a very rich diversity. Full resolution of the origin of cBasmati may require further investigation given the vast diversity it encompasses [78, 79]. SNAPPNET provides here a consistent and convincing set of results. Its integration in BEAST may provide easier applicability than previous methods, potentially making it a method of choice to expand analysis of complex pictures generated by crop evolution and adaptation. Further applicability advantages may come from the fact that SNAPPNET can be used to compute the likelihoods of a set of networks of interest, and then to penalize more complex models with the AIC [80] and BIC [81] criteria.

In the future, in order to handle more sites in practice, the MSNC model should be extended to allow recombination events between loci. Recall that we have limited our rice study to 12,000 markers sampled along the genome because our model assumes independence between sampled sites, as does also SNAPP's model, from which we inherit. As mentioned in the review of [38], in order to model recombination properly, the study of gene networks within species networks is an area for future research. A possibility would be to exploit previous work on Ancestral Recombination Graphs (see for instance [82]).

Another important research topic for MCMC inference of phylogenetic networks is the question of how to properly assess the autocorrelation between the topologies of the sampled networks, or, in other words, how to estimate the effective sample size (ESS) of the sampled topologies. Indeed, a large ESS for continuous parameters in a phylogenetic model does not necessarily imply a large ESS for the sampled topologies. Methods to estimate the ESS of a sample of tree topologies have been recently proposed [70]. They rely on measures of the distance between pairs of trees in the sample—which enable to assess autocorrelation—or on translating tree topologies into numbers (e.g., the distance from a focal tree), which are then treated as continuous parameters—for which an ESS can then be computed using standard approaches. These methods to estimate topological ESS can be in principle adapted to networks. However some research will be needed for this, as standard tree metrics (e.g. the Robinson-Foulds distance [83] or the path-lengths difference [84]) do not have unique, easy to compute, natural extensions for networks (see [38] for a discussion on this). In the present work, different MCMC replicates led to consistent results, but we have not attempted to evaluate

autocorrelation for the sampled topologies and/or their ESS. This is a limitation of all Bayesian approaches for network inference proposed so far [51, 53, 54].

Related to the issue above, it would be useful to conduct an in-depth investigation of the efficiency of the MCMC operators for the exploration of network topology space. In this work, we rely on the operators by Zhang et al. [53], who identified this as a major bottleneck of their approach (but they also had operators to change the gene tree embeddings, a feature that we do not need here). Although some important progress has been made in the last 20 years [85], in 2004 Felsenstein aptly wrote (speaking about trees): "At the moment the choice of a good proposal distribution involves the burning of incense, casting of chicken bones, magical incantations and invoking the opinions of more prestigious colleagues" [14]. Since network space is significantly more complex than tree space, it is easy to predict that this topic will keep researchers busy for a long time. A good starting point to address convergence issues in SNAPP-NET would be to integrate it to the new BEAST 2 package COUPLED MCMC [86], which tackles local optima issues thanks to heated chains.

Also note that in this work we limited our experiments to relatively simple networks, with few reticulations and few species (leaves). While the number of reticulations represents a strong limitation of all existing Bayesian approaches, the number of species is a much weaker limiting factor. Networks over more species can already be inferred by SNAPPNET and related approaches, but MCMC inference for such networks will then necessitate much more complex downstream analyses than the ones used here. For example, the posterior probability of any single network topology will be very small, and thus it will be much more interesting to look at the probability of individual splits, or to develop a network analog of consensus trees. These are not simple tasks, because all the underlying algorithmic problems (checking the presence of a split/clade in a network, or that of a subtree etc.) are computationally hard to solve on large networks [87].

Last, it would be interesting to study the identifiability of the model underlying SNAPPNET. For example, it is easy to see that if only one lineage is sampled from a given species at each locus, then the population size $\theta$ of that species is non-identifiable (because no coalescence can ever occur in it, and thus the likelihood does not depend on $\theta$). Similarly, if only one lineage is sampled below a reticulation node, then the height of that node is non-identifiable [41, 61]. Intuitively, the more lineages can co-exist in a part of the species network, the more information there will be for the reconstruction of that part of the network. These aspects should be further investigated in future works.

Many methodological questions on Bayesian inference of phylogenetic networks remain open. The present work focused on the efficient calculation of likelihood for a single network, which is the key component of any Bayesian approach. At the end of their paper, the authors of `MCMC_BiMarkers` [54] concluded by mentioning that "An important direction for future research is improving the computational requirements of the method to scale up to data sets with many taxa". Our present work is a first answer to this demand.

## Supporting information

**S1 Text. Supplementary material for the manuscript. Fig A:** Density probabilities for 5-tips networks, simulated with a prior corresponding to a birth hybridization process with parameters $d = 10$, $r = 1/2$ and $\tau_0 = 0.1$, using the SPECIESNETWORK package [53]. The figure is obtained for 10,000 replicates. The means are given by the dashed vertical lines. **Fig B:** Density probabilities for 5-tips networks with at most two reticulations, simulated with a prior corresponding to a birth hybridization process with parameters $d = 10$, $r = 1/2$ and $\tau_0 = 0.1$, using the SPECIES-NETWORK package [53]. Figures are drawn for the 4,377 cases in 10,000 where the network had

at most two reticulations. The means are given by the dashed vertical lines. **Fig C:** Density probabilities regarding the 5-tips network with a maximum of 3 reticulations, simulated under the birth hybridization process ($d = 10$, $r = 1/2$, $\tau_0 = 0.1$, 5,837 replicates), using the SpeciesNetwork package [53]. The means are given by the dashed vertical lines. **Fig D:** Estimated node heights of network B. 10,000 sites are considered and 2 lineages per species. Constant sites are included in the analysis, and the estimated heights are based on the 12 replicates (over 14 replicates) for which network B was recovered by SnappNet (criterion ESS > 200; $\theta \sim \Gamma(1, 200)$, $d \sim \mathcal{E}(0.1)$, $r \sim \text{Beta}(1, 1)$, $\tau_0 \sim \mathcal{E}(10)$ for the priors, number of reticulations bounded by 3 when exploring the network space). Heights are measured in units of expected number of mutations per site. True values are given by the dashed horizontal lines. The initials MRCA stand for "Most Recent Common Ancestor". **Fig E:** Estimated population sizes $\theta$ for each branch of network B. Same framework as Figure D in S1 Text. True values are given by the dashed horizontal lines. The initials MRCA stand for "Most Recent Common Ancestor". **Fig F:** Same framework as Figure E in S1 Text. **Fig G:** Estimated node heights of network C as a function of the number of sites. Same experiment as in Table 2 of the main manuscript: 1 lineage in species O, A and D, and 4 lineages in species B and C. The estimated heights are based on the replicates for which network C was recovered by SnappNet. True values are given by the dashed horizontal lines. The initials MRCA stand for "Most Recent Common Ancestor". **Fig H:** Estimated height and length for network A, as a function of the number of sites. Heights and lengths are measured in units of expected number of mutations per site. True values are given by the dashed horizontal lines. Two lineages per species were simulated. Only polymorphic sites are included in the analysis, and 20 replicates are considered for each simulation set up (criterion ESS > 200 for m = 1,000 and m = 10,000, and criterion ESS > 100 for m = 100,000; $\theta \sim \Gamma(1, 200)$, $d \sim \mathcal{E}(0.1)$, $r \sim \text{Beta}(1, 1)$, $\tau_0 \sim \mathcal{E}(10)$ for the priors, number of reticulations bounded by 2 when exploring the network space). Same framework as in Fig 10 of the main paper, except that only polymorphic sites are taken into account. **Fig I:** Estimated inheritance probability and instantaneous rates for network A, as a function of the number of sites. True values are given by the dashed horizontal lines. Same framework as in Fig 11 of the main paper, except that only polymorphic sites are taken into account. **Fig J:** Estimated node heights of network A, as a function of the number of sites. Heights are measured in units of expected number of mutations per site. True values are given by the dashed horizontal lines. Same framework as in Fig 12 of the main paper, except that only polymorphic sites are taken into account. The initials MRCA stand for "Most Recent Common Ancestor". **Fig K:** Estimated population sizes $\theta$ for each branch of network A, as a function of the number of sites. True values are given by the dashed horizontal lines. Same framework as in Fig 13 of the main paper, except that only polymorphic sites are taken into account. The initials MRCA stand for "Most Recent Common Ancestor". **Fig L:** Experiments on Network A and based only on polymorphic sites. Same framework as in Figures H and I in S1 Text, except that the correction factor is not used in the calculations (criterion ESS > 200 in all cases). **Fig M:** Summary of rice molecular diversity used for selecting our sample of rice cultivated varieties and wild types. (A) unweighted neighbour joining (UWNJ) tree reflecting dissimilarities among 899 accessions based on 2.48 million SNPs as described in [73]; the accessions are colored according to their classification into wild population types or cultivar groups. (B, C) UWNJ tree using the same data for the 24 accessions we selected for assessing SnappNet performance, and showing their accessions number (B) and their country of origin (C); the colors are as in A. **Fig N:** Trace plots obtained according to the Tracer software when data set 1 was analyzed with SnappNet. (a) and (b) refer to the first sampling of 12 kSNPs along the whole genome, whereas (c) and (d) focus on the second sampling. Two chains were considered for each sampling. **Fig O:**

Birth-hybridisation model with speciation rate 20 and hybridisation rate 1 (mean number of reticulations close to zero) and a normal prior with mean 0.1 and standard deviation of 0.01 on the origin height. We plot the simulated networks (orange) against the sampled networks (blue) summarising the networks under: (a) Number of reticulations (b) Time until first reticulation (c) Height of the network (d) Length of the network. **Fig P:** Birth-hybridisation model with speciation rate 20 and hybridisation rate 2 (mean number of reticulations close to one) and normal prior with mean 0.1 and standard deviation of 0.01 on the origin height. We plot the simulated networks (orange) against the sampled networks (blue) summarising the networks under: (a) Number of reticulations (b) Time until first reticulation (c) Height of the network (d) Length of the network. **Fig Q:** Birth-hybridisation model with speciation rate 20 and hybridisation rate 3 (mean number of reticulations close to two) and normal prior with mean 0.1 and standard deviation of 0.01 on the origin height. We plot the simulated networks (orange) against the sampled networks (blue) summarising the networks under: (a) Number of reticulations (b) Time until first reticulation (c) Height of the network (d) Length of the network. **Fig R:** Birth-hybridisation model with speciation rate 20 and hybridisation rate 1 (mean number of reticulations close to zero) and an exponential prior with mean 0.1 on the origin height. We plot the simulated networks (orange) against the sampled networks (blue) summarising the networks under: (a) Number of reticulations (b) Time until first reticulation (c) Height of the network (d) Length of the network. **Fig S:** Birth-hybridisation model with speciation rate 20 and hybridisation rate 2 (mean number of reticulations close to one) and an exponential prior with mean 0.1 on the origin height. We plot the simulated networks (orange) against the sampled networks (blue) summarising the networks under: (a) Number of reticulations (b) Time until first reticulation (c) Height of the network (d) Length of the network. **Fig T:** Birth-hybridisation model with speciation rate 20 and hybridisation rate 3 (mean number of reticulations close to two) and an exponential prior with mean 0.1 on the origin height. We plot the simulated networks (orange) against the sampled networks (blue) summarising the networks under: (a) Number of reticulations (b) Time until first reticulation (c) Height of the network (d) Length of the network. **Fig U:** Summary distributions of all chains with correct population size priors (chain numbers 1,2,9,10,17,18) given data simulated from network A. We summarize the MCMC chains by combining them, that is: Chains 1 and 2 are indicated by the blue line (mean reticulations close to zero); Chains 9 and 10 are indicated by the orange line (mean reticulations close to one); Chains 17 and 18 are indicated by the green line (mean reticulations close to two); We plot the following distributions (a) Likelihood (b) Prior (c) Network height (d) Network length. Note that network height and network length used to simulate data are indicated by red lines. **Fig V:** Summary distributions of all chains with incorrect population size priors Gamma(1,20) (chain numbers 3,4,11,12,19,20) given data simulated from network A. We summarize the MCMC chains by combining them, that is: Chains 3 and 4 are indicated by the blue line (mean reticulations close to zero); Chains 11 and 12 are indicated by the orange line (mean reticulations close to one); Chains 19 and 20 are indicated by the green line (mean reticulations close to two); We plot the following distributions (a) Likelihood (b) Prior (c) Network height (d) Network length. Note that network height and network length used to simulate data are indicated by red lines. **Fig W:** Summary distributions of all chains with correct population size priors (chain numbers 1,2,9,10,17,18 given data simulated under network B. We summarize the MCMC chains by combining them, that is: Chains 1 and 2 are indicated by the blue line (mean reticulations close to zero); Chains 9 and 10 are indicated by the orange line (mean reticulations close to one); Chains 17 and 18 are indicated by the green line (mean reticulations close to two); We plot the following distributions (a) Likelihood (b) Prior (c) Network height (d) Network length. Note that network height and network

length used to simulate data are indicated by red lines. **Fig X:** Summary distributions of all chains with incorrect population size priors (chain numbers 3,4,7,8,11,12) given data simulated from network B. We summarize the MCMC chains by combining them, that is: Chains 3 and 4 are indicated by blue line (mean reticulations close to zero); Chains 7 and 8 are indicated by orange line (mean reticulations close to one); Chains 11 and 12 are indicated by green line (mean reticulations close to two); We plot the following distributions (a) Likelihood (b) Prior (c) Network height (d) Network length. Note that network height and network length used to simulate data are indicated by red lines. **Fig Y:** In this we figure we plot summary distributions of all chains with incorrect population size priors Gamma(1,20) (chain numbers 5,6,13,14,21,22) given data simulated from Network B. We summarize the MCMC chains by combining them, that is: Chains 5 and 6 are indicated by blue line (mean reticulations close to zero); Chains 13 and 14 are indicated by orange line (mean reticulations close to one); Chains 21 and 22 are indicated by green line (mean reticulations close to two); We plot the following distributions (a) Likelihood (b) Prior (c) Network height (d) Network length. Note that network height and network length used to simulate data are indicated by red lines. **Fig Z:** In this we figure we plot summary distributions of all chains with incorrect population size priors (chain numbers 7,8,15,16,23,24) given data simulated from network B. We summarize the MCMC chains by combining them, that is: Chain 7 and 8 are indicated by blue line (mean reticulations close to zero); Chain 15 and 16 is indicated by orange line (mean reticulations close to one); Chain 23 and 24 are indicated by green line (mean reticulations close to two); We plot the following distributions (a) Likelihood (b) Prior (c) Network height (d) Network length. Note that network height and network length used to simulate data are indicated by red lines. **Table A:** Table linked to Table 1 of the main manuscript. Trees inferred by SNAPPNET when m = 1,000 sites were considered. **Table B:** Average posterior probability (PP) of the topology of network C obtained by running `MCMC_BiMarkers` on data simulated from network C. Same as Table 3 of the main manuscript except that $12 \times 10^6$ iterations are considered, and only one lineage is sampled in hybrid species B and C. $\overline{\text{ESS}}$ is the average ESS over the different replicates, and SE stands for the sampler efficiency. **Table C:** Description of the 24 rice varieties considered in our study. These varieties are either representative cultivars spanning the four main rice subpopulations (Indica, Japonica, *circum* Aus and *circum* Basmati), or wild types (Or1I, Or1A, Or3). **Table D:** Data set 1, that includes only one variety per subpopulation. These varieties were chosen from Table C in S1 Text. **Table E:** Data sets 2 and 3, that include two varieties per subpopulation. These varieties were chosen from Table C in S1 Text. **Table F:** Informations obtained according to the Tracer software, when data set 1 was analyzed with SNAPPNET. Two different samplings of 12 kSNPs were considered, and also two chains for each sampling. **Table G:** BH(birth rate, hybridisation rate) refers to the birth-hybridisation process of Zhang et al. with the specified birth and hybridisation rates. For data simulated with network A, only chains 1,2,3,4,9,10,11,12,17,18,19,20 were run. We indicate the mean number of reticulation for the Birth-Hybridization model given an exponential prior with mean 0.1 on network origin. Note that we only used the exponential prior in the experiment in Section 8.2 of S1 Text. **Table H:** MCMC summary statistics for network A (correct population size priors). **Table I:** MCMC summary statistics for network A (incorrect priors). **Table J:** MCMC summary statistics for Network B (correct population size priors). **Table K:** MCMC summary statistics for Network B (incorrect population size priors Gamma(1,20)). **Table L:** MCMC summary statistics for Network B (incorrect population size priors Gamma(1,1000)). **Table M:** MCMC summary statistics for Network B (incorrect population size priors Gamma(1,2000)). **Table N:** MCMC acceptance rates for Network B (correct population size priors). **Table O:** MCMC acceptance rates for Network B (incorrect population size priors

$\Gamma(1, 1000)$). **Table P:** MCMC acceptance rates for Network B (incorrect population size priors $\Gamma(1, 2000)$).
(PDF)

## Acknowledgments

VB thanks University Montpellier for a 6-month sabbatical period.

## Author Contributions

**Conceptualization:** Fabio Pardi.

**Data curation:** Vincent Berry, João D. Santos, Wensheng Wang, Jean-Christophe Glaszmann.

**Formal analysis:** Charles-Elie Rabier, Vincent Berry, Fabio Pardi, Celine Scornavacca.

**Funding acquisition:** Vincent Berry, Celine Scornavacca.

**Investigation:** Charles-Elie Rabier, Vincent Berry, Fabio Pardi, Celine Scornavacca.

**Methodology:** Charles-Elie Rabier, Vincent Berry, Fabio Pardi, Celine Scornavacca.

**Project administration:** Vincent Berry, Celine Scornavacca.

**Resources:** Charles-Elie Rabier, Vincent Berry, Celine Scornavacca.

**Software:** Charles-Elie Rabier, Vincent Berry.

**Supervision:** Vincent Berry, Jean-Christophe Glaszmann, Fabio Pardi, Celine Scornavacca.

**Validation:** Charles-Elie Rabier, Vincent Berry, Marnus Stoltz, Jean-Christophe Glaszmann, Fabio Pardi, Celine Scornavacca.

**Visualization:** Charles-Elie Rabier, Marnus Stoltz, Fabio Pardi, Celine Scornavacca.

**Writing – original draft:** Charles-Elie Rabier, Vincent Berry, Jean-Christophe Glaszmann, Fabio Pardi, Celine Scornavacca.

**Writing – review & editing:** Charles-Elie Rabier, Vincent Berry, Marnus Stoltz, Jean-Christophe Glaszmann, Fabio Pardi, Celine Scornavacca.

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
