## [Decision Letter · Decision Letter 0]

3 Feb 2021

Dear Dr Rabier,

Thank you very much for submitting your manuscript "On the inference of complex phylogenetic networks by Markov Chain Monte-Carlo" for consideration at PLOS Computational Biology.

As with all papers reviewed by the journal, your manuscript was reviewed by members of the editorial board and by several independent reviewers. In light of the reviews (below this email), we would like to invite the resubmission of a significantly-revised version that takes into account the reviewers' comments.

Particular attention should be paid to overhauling the "real data" analysis part of the manuscript (the reviewers were quite disappointed by the current one), and to improving the depth and clarity (although opinions differed there) of the methodological exposition.

We cannot make any decision about publication until we have seen the revised manuscript and your response to the reviewers' comments. Your revised manuscript is also likely to be sent to reviewers for further evaluation.

Sincerely,

Sergei L. Kosakovsky Pond, PhD

Associate Editor

PLOS Computational Biology

William Noble

Deputy Editor

PLOS Computational Biology

Reviewer's Responses to Questions

**Comments to the Authors:**

Reviewer #1: The authors presented an extremely well-written manuscript with the extension of SNAPP to networks (SnappNet) which is similar in principle to the work on Zhu et al (2018) but with two key differences: the way in which the likelihood is computed and the priors. These two differences have a huge impact on the method's computational speed and convergence.

I want to thank the authors for the high quality writing and figures. It was a joy to read it!

# Major comments

- I would appreciate a bit more details or examples (maybe figures?) to better explain two concepts: "population interfaces" and "incomparable population interfaces". The whole section on partial likelihoods relies on these two concepts and I found them very very hard to understand (perhaps given my inexperience with Snapp to begin with). For readers like me that are not familiar with these concepts, a bit more explanation will make this section much easier to digest

- Did you make experiments on how the prior affects convergence and computational speed? Given that the prior is an important difference with the work in Zhu et al (and that we know that it has some weight on the mixing of the chain), it would be interesting to compare how the network priors behave in terms of convergence and speed, not just accuracy

- It seems that the tests on the accuracy of the numerical parameters is done only on Network A (unless I missed this for Network B and C). I think that testing the accuracy of estimated branch lengths and network height is very important for complex networks like B and C, so I wish something could be said about this. In addition, for complex networks it seems that the number of hybridizations was bounded at the true number of hybridizations which is a good initial test, but I would be interested to know how the method performs when the prior is more lax and allows for more complex networks. What networks do you end up visiting frequently if the prior allows for more hybridizations than the true number? I know that this type of tests could easily get out of hand, but doing one or two simulations in which the prior is not bounding the number of hybridizations could provide information on accuracy and convergence under more realistic settings. In fact, perhaps this could be one of the things happening on the real data, that the number of hybridizations is not correctly guessed for the prior

- It seems that Network C(3) and C(4) were only used to test CPU time and memory, which is indeed important, but if the algorithms were run anyway, it would be nice to see the accuracy of the method for these more complex networks

- The real data analysis was the only disappointing part of the manuscript. The fact that the algorithm could not search adequately the space of networks and that 10 networks had to be arbitrarily chosen for comparison was very disappointing. It was also not explained what criteria was used to choose the 10 networks? I understand that real data will never behave like simulated data, but if the current algorithm is not applicable to this dataset (that appears to be quite similar, at least in terms of sample sizes to the simulated data), which real data could be used for this method? Also, do the authors have a sense for what made the real data so difficult to analyze? Because the sample size looks quite manageable (12,000 sites and 4 taxa). I think that a thorough investigation into why this dataset could not be analyzed with this method would provide users with more information regarding the limitations of the method

# Minor comments

- Line 62: "networks that can lead to observe such trees" sounds like it is referring to the concept of displayed tree which I think is not accurate in this section related to likelihood methods

- Line 228: I would add a "in this case" to the sentence of "Note that N_{\\underline{x}} is not random" because it is not random in general, right?

- Line 397: how are the "starting points" for the MCMC chosen?

- Line 427: why not say "estimated posterior probability" instead of "ratio", am I missing something?

- Lines 450-463: It is unclear the difference between the two optimizations. In the first case, you mention that only 9 operators were used (and ignored the 7 related to network updates) but in the second case you mention that the network topology was fixed. Isn't that the same thing as the first optimization?

- Lines 471-473: I don't understand the evaluation criterion, can you clarify?

- Line 502: In the cases where a network with only 1 reticulation was recovered, was this at least a subnetwork of the true network?

- Table 1: I was intrigued that on the third row (less ILS) the accuracy in Network A decreased from 100% to 80% as you increased more sites. Do the authors have an idea why this could be happening?

- Table 2: When the number of lineages are varied for B and C, you choose the same number of lineages for both, right? Either B and C have only one lineage, or they both have 4, correct?

- Table 7: there is a missing reference to a Figure

- Figure 13: what do the authors refer to "Network length". I am sure this most be defined in the text, but I did not find it

- Figure 14 (and others): I would repeat the caption instead of writing "Same as Figure 10" for readability

Reviewer #2: Rabier et al. have developed an elegent and potentially revolutionary algorithm to compute the likelihood of biallelic data given a species network. Not only does their manuscript present the algorithm in sufficient clarity and detail, but several analyses of real and simulated data sets demonstrates its computational performance and general utility of biallelic inference of species networks.

I only have a few comments regarding the manuscript.

1. How was the correctness of the likelihood implementation validated? It would make sense to compare the likelihood computed by the algorithm of Zhu et al. with SnappNet for a random species network, disagreement would suggest an error in one or both implementations. Or it could possibly be computed manually for a simple network and compared with either implementation.

2. The language in the introduction is a little oversimplified, especially the sentence "Mechanisms at stake lead a genome to have different parent species - in contrast with the simpler image that depicts a genome as being vertically inherited with modifications from a single ancestral genome." Under the multispecies coalescent a species can have a single parent, but inherited multiple ancestral genomes at the top of the branch. This is followed an overview of different kinds of "mechanisms leading to mix genome contents" (grammatically incorrect, maybe this should be "mechanisms leading to mixed inheritance"?), including HGT in prokaryotes. I think it would be better to remove this or make it clear that the model implemented here isn't useful to model HGT in prokaryotes, since it relies on the free recombination between sites facilitated by sexual reproduction.

3. A limitation inherent to inference of species networks using biallelic markers is the apparent non-identifiability of reticulation node heights (between upper and lower bounds). This is shown in section 2.1 of Cao, Zhen, et al. "Practical aspects of phylogenetic network analysis using phylonet." bioRxiv (2019). The authors may consider including this in the discussion, as it seemingly the case that precise inference of reticulation node heights requires other kinds of methods (e.g. multilocus methods or coalescent HMMs). However this does not diminish the utility of Rabier et al.'s method for the inference of the species network topology or speciation times.

I enjoyed reading the manuscript and am looking forward to seeing the method used in practice.

Regards,

Huw A. Ogilvie

Reviewer #3: The authors developed a new Bayesian method, SnappNet for phylogenetic

network inference. The authors use a faster new method based on joint

conditional probabilities to compute likelihoods of phylogenetic

networks resulting in faster analysis by orders of magnitude and with an

order of magnitude less memory usage than the recent MCMCBiMarkers

method for inferring networks.

This work was inspired by the authors of MCMCBiMarkers noting that it is

necessary to improve the speed of such methods. The authors leverage the

improved speed to consider more complex evolutionary scenarios with an

example of inferring a new evolutionary scenario on rice genomic data that

is compatible with available evidence.

This is an important development and will be of great interest to the

readers of PLOS Computational Biology. The new fast likelihood computation

method for networks is novel and useful. However, the analysis of the

inferred networks needs major revisions. The treatment of

the complicated topic of convergence in a Bayesian analysis needs to be

substantially expanded. The authors seem to consider the inferred posterior

probability of a network as an accuracy measure and there are major

questions about whether the method is correctly inferring the expected

network or simply stuck at that network. Finally, The paper needs a much more

careful analysis of the differences between SnappNet and MCMCBiMarkers

results in order to support their accuracy claims.

Specific comments follow

Page 12 - MCMC Operators

This section is woefully short considering the great importance of MCMC

operators on the convergence and accuracy of a Bayesian analysis. What

to the most important operators do? How are the operators tailored to

work with a network as opposed to a tree? Which operators are most used

and what are the acceptance rates? Is there any possiblity that the

operators have a bias towards or away from certain types of network

shapes? If these questions have not been answered in specific detail yet

then they should be included in the discussion and mentioned as

important future work. I see that the authors cite SpeciesNetwork as the

origin of the network topology operators but it is important to mention

such concerns at the very least, particularly given that you are

combining them with new parameters and new operators.

Page 13 - Phylogenetic networks studied

Why did you study these three networks? Are they representative of

important type of networks or biologically important in some way?

Are there any caveats to focusing on these networks or areas of future

work that should be studied on larger or more complex networks?

Page 13 - Bayesian Analysis

I see in the commands in the supplemental material that you sampled every 1000

iterations of the MCMC. This is worth mentioning in the paper proper

along with an explanation that this is to prevent autocorrelation and

part of the reason that computational complexity of likelihood

calculations is important.

Why did you use a single starting tree or network for each MCMC

analysis instead of a random starting point? This is irregular in

MCMC analysis and makes it difficult to generalize your results.

This needs more explanation and justification.

Page 14 - MCMC convergence

More detail is needed on what parameters the ESS was measured on. The linked

BEAST tutorial mentions that no topological ESS measures are included by

default so I am not sure what ESS was measured and whether any tests of

topological convergence were performed.

In general the explanation of convergence needs to be expanded.

Convergence is a complicated yet important topic and a threshold of ESS

is most certainly not enough by itself to assume convergence, that is

an ESS of each important parameter is a necessary but not sufficient

requirement to assume convergence.

Importantly, convergence of treelength or network length does not imply the

convergence of topology. ESS measures for tree topology have been

recently developed and the authors should note whether they have

considered similar measures for network topology ESS or indicate that

this is an important area for future work. If a topology ESS is not used

(or possible) then at the very least multiple MCMC replicates should be tested and the

results compared to determine if similar topologies, split frequencies,

or network edges are explored before assuming convergence.

page 15 - Study of networks A and B

Table 1 should be closer to its usage.

page 15-16 Ability to recover the network topology

I am very confused about what your measure of accuracy is supposed to be

for the correct network topology here and in Table 1. From the text it

seems that you are reporting the percentage of total topologies sampled

that matched the target/starting topology. However, in a Bayesian MCMC

analysis the percentage of samples should be proportional to the

posterior probability of that topology, not some measure of accuracy. We

should expect the true network to be the most frequently sampled one if the

model is correct but values approaching 100% do not seem worth

applauding as they may indicate lackluster topology proposals and that the

analysis is stuck at the initial network, or alternatively that the

analysis is relatively trivial.

Similarly, the accuracy on network B has not "decreased slightly to

81.25%" but rather the posterior probability of network B is 81.25%

indicating that other scenarios are possible.

A better analysis should probably be based on some idea of consensus networks, as

an analog to consensus trees, and whether the MCMC analysis can

efficiently sample from the probability space and recover that

consensus. Some form of consensus analysis could be used to determine the

posterior probability of individual splits or network edges and I

suspect, as with trees, this will be necessary for studying larger

or more complex scenarios where the true network may have a relatively

small probability.

Page 18 - It's difficult to justify the conclusion that MCMCBiMarkers is not

performing well if MCMCBiMarkers was not run to convergence. Testing

with larger ESS values is not the same as running it until the MCMC has

fully converged.

Page 22 - You mention here that on the real datasets with free

exploration of the topology space the analysis frequently got stuck in

local optima. As with the simulated data this suggests that new methods

are needed to sample from networks.

You explain here that you focused on direct analysis of 10 specific

topologies by computing their likelihoods and using AIC and BIC

penalties. This is one solution but it needs to be more carefully

explained to the reader what this means. You should also explain

carefully here that this is not a Bayesian analysis as you have moved

from posterior probablity to likelihood comparisons. The networks can be

compared against each other but it is not possible to consider their

probability with this type of analysis and it is possible that some

other network you do not consider or specific network features have a

higher probability than any of these 10 networks.

Page 27 - the analysis of the candidate networks is interesting in that

it discusses biologically relevant factors and the penalized likelihood

ranking of the networks but I think some of the conclusions here are too strong

based on just this evidence, such as suggesting that specific networks should

be studied more closely based on this analysis or that

specific networks should be discounted because their results would be

surprising biologically.

The mention of heated chains for future work seems out of place here in the results

section and should likely be in the discussion or conclusion.

Page 29 - the comparison of SnappNet and MCMCBiMarkers "accuracy" on

network C again seems to be without context. More justification is

needed to prove that the low accuracy of the expected network is a

detriment. For example, I suggest discussing the posterior probability

from each method that the network has a given number of reticulations

to better make your claim that MCMCBiMarkers is discounting these scenarios.

**Have all data underlying the figures and results presented in the manuscript been provided?**

Reviewer #1: Yes

Reviewer #2: Yes

Reviewer #3: Yes

PLOS authors have the option to publish the peer review history of their article (what does this mean?). If published, this will include your full peer review and any attached files.

Reviewer #1: No

Reviewer #2: **Yes: **Huw A. Ogilvie

Reviewer #3: No
---

## [Decision Letter · Decision Letter 1]

13 Jul 2021

Dear Dr Rabier,

We are pleased to inform you that your manuscript 'On the inference of complex phylogenetic networks by Markov Chain Monte-Carlo' has been provisionally accepted for publication in PLOS Computational Biology.

Best regards,

Sergei L. Kosakovsky Pond, PhD

Associate Editor

PLOS Computational Biology

William Noble

Deputy Editor

PLOS Computational Biology

Prior to submitting files for production, please address the remaining minor comments raised by Rev 1.

Reviewer's Responses to Questions

**Comments to the Authors:**

Reviewer #1: I want to thank the authors for addressing all my comments and questions in a very effective manner. The description of the model is much easier to follow thanks to the additions, and the simulations are more convincing.

My only minor comments relate to the real data analysis. I was excited that the chains were run to estimate the networks (as opposed to fixing the networks as in the initial version).

# Minor comments

- Lines 719-720: can you add the reference to where we can find these convergence checks in the text (or maybe they are in the supp mat)? I could not find the trace plots or how where the posterior distributions compared

- Line 724: you mention that all networks sampled had the same topology. Is this true also for the burnin period? (Side note: not sure if burnin info was included in the text). How was the starting point of the chain determined? Where different starting points used?

- Paragraph starting on Line 729: do you also get the chain containing only one topology for data set 2 (as for data set 1)? How was the starting point of the chain determined? Where different starting points used?

- Figure 18: Could the second and third topology be flipped so that the taxa is ordered in the same way as the first topology? This is just to facilitate comparison. I suggest flipping the 2nd and 3rd (rather than only the 1st topology) because the 1st topology agrees with the other figures

- Was the real data run with PhyloNet too? Or was it prohibited?

Reviewer #3: The authors developed a new Bayesian method, SnappNet for phylogenetic

network inference. The authors use a faster new method based on joint

conditional probabilities to compute likelihoods of phylogenetic

networks resulting in faster analysis by orders of magnitude and with an

order of magnitude less memory usage than the recent MCMCBiMarkers

method for inferring networks.

This work was inspired by the authors of MCMCBiMarkers noting that it is

necessary to improve the speed of such methods. The authors leverage the

improved speed to consider more complex evolutionary scenarios with an

example of inferring a new evolutionary scenario on rice genomic data that

is compatible with available evidence.

This is an important development and will be of great interest to the

readers of PLOS Computational Biology. The new fast likelihood computation

method for networks is novel and useful.

The authors have satisfied all of my concerns with the original submission.

They now carefully explain how the model can be used with an explanation of

convergence and a much better explanation of the limitations of this work.

The expanded analysis of rice genomic data is much more thorough and

shows the potential of their method. The authors also now explain in greater

detail what new advances are needed beyond faster likelihood calculations

to accurately infer larger and more complex networks.

**Have the authors made all data and (if applicable) computational code underlying the findings in their manuscript fully available?**

Reviewer #1: Yes

Reviewer #3: Yes

PLOS authors have the option to publish the peer review history of their article (what does this mean?). If published, this will include your full peer review and any attached files.

Reviewer #1: No

Reviewer #3: No

---

## [Editor Report · Acceptance letter]

27 Aug 2021

PCOMPBIOL-D-20-01799R1 

On the inference of complex phylogenetic networks by Markov Chain Monte-Carlo

Dear Dr Rabier,

I am pleased to inform you that your manuscript has been formally accepted for publication in PLOS Computational Biology. Your manuscript is now with our production department and you will be notified of the publication date in due course.

With kind regards,

Livia Horvath
